# Ballistic supercavitating nanoparticles driven by single Gaussian beam optical pushing and pulling forces

Eungkyu Lee [1], Dezhao Huang[1] & Tengfei Luo [1,2,3]✉

Directed high-speed motion of nanoscale objects in fluids can have a wide range of applications like molecular machinery, nano robotics, and material assembly. Here, we report ballistic plasmonic Au nanoparticle (NP) swimmers with unprecedented speeds (~336,000 μm s$^{-1}$) realized by not only optical pushing but also pulling forces from a single Gaussian laser beam. Both the optical pulling and high speeds are made possible by a unique NP-laser interaction. The Au NP excited by the laser at the surface plasmon resonance peak can generate a nanoscale bubble, which can encapsulate the NP (i.e., supercavitation) to create a virtually frictionless environment for it to move, like the Leidenfrost effect. Certain NP-in-bubble configurations can lead to the optical pulling of NP against the photon stream. The demonstrated ultra-fast, light-driven NP movement may benefit a wide range of nano- and bio-applications and provide new insights to the field of optical pulling force.

[1] Department of Aerospace and Mechanical Engineering, University of Notre Dame, Notre Dame, IN 46556, USA. [2] Department of Chemical and Biomolecular Engineering, University of Notre Dame, Notre Dame, IN 46556, USA. [3] Center for Sustainable Energy of Notre Dame (ND Energy), University of Notre Dame, Notre Dame, IN 46556, USA. ✉email: tluo@nd.edu

Speed is important to the functionality of nanoswimmers/microswimmers in fluids because it determines the efficiency of swimmers[1–4]. Directed movement is also important as the swimmers are usually tasked to reach a target. However, directed movement and high speed in the nanoscale are rarely compatible[5–7]. In laminar fluid flow, enabling fast-moving swimmers requires large propulsion force, as it needs to counter the drag force which is proportional to the speed[8]. Usually, nanoswimmers/microswimmers driven by on-board propulsion (e.g., bubble repellence[5–7]) exhibit higher speeds compared to other mechanisms (e.g., mechanical motion[9,10], pressure[11] and thermal[12,13], gradients, chemical phoresies[14–16], and optical force[17,18]). The highest speed of bubble-repelled swimmers is $\sim 10^5$ body-length s$^{-1}$ (see ref. [7]), where the unit (the speed to body-length ratio) is commonly used to allow a fair comparison between swimmers with different dimensions. However, it is very difficult to control their moving directions because of the random nature of the on-board repellence force.

Optical forces on an object due to the exchange of photon momentum with the object[17–27] can move nanoswimmers/microswimmers along the beam direction. Light propulsion has unique advantages like wireless control, high spatial and temporal precision, and instant response[28]. Usually, light applies a pushing force on an object in the light propagating direction, which has worked together with the optical gradient force to enable the optical tweezer effect[17,19,20]. Objects with high scattering efficiency like plasmonic NPs can enable stronger optical pushing forces[19,20], but the achieved highest swimmer speed is merely $\sim 10^3$ body-length s$^{-1}$ (see ref. [17]), since the optical forces are very weak ($10^{-12}$ to $10^{-14}$ N).

Light may also pull an object in specific optical conditions[18,21–27]. For example, two planewaves irradiating mirror-symmetrically with a large incidence angle towards an object can lead to a strongly focused forward photon scattering momentum, resulting in a net "negative" optical force[22–24]. For a spheroidal object much larger than the wavelength of the incident light and placed at a heterogenous dielectric interface, the object can be pulled by the light due to an increase in the photon linear momentum across the dielectric interface[18]. However, the achieved moving speed is only $\sim 0.5$ body-length s$^{-1}$, and the same strategy may not be applicable to objects in homogenous media, which are the cases for most nanoswimmers/microswimmers. Theoretically, an object in homogenous medium can also be pulled by a single planewave if the object has certain unique optical configurations to enable either optical gain[25,26] or near-field electromagnetic coupling between dielectric-metallic NP dimers[27]. However, the experimental demonstration in homogenous media is still lacking.

In this work, we observe optical pushing and pulling of Au NP swimmers exhibiting extremely high speeds ($>10^6$ body-length s$^{-1}$) in water, and we show that the unusual optical pulling and high-speed movement are only made possible by the nano-bubbles generated around the laser-excited plasmonic NPs, which lead to supercavitation.

## Results

### Ballistic motion of supercavitating nanoparticle.

The experimental setup is shown in Fig. 1a. We disperse Au core–shell NPs made of a silica core ($\sim 100$ nm) and an Au shell ($\sim 10$ nm) in deionized water. A femtosecond pulsed laser (linear-polarized Gaussian beam) with a repetition rate of 80.7 MHz and a wavelength of 800 nm is focused in the NP-water suspension using a 20× objective lens (numerical aperture $\sim 0.42$). At the focal plane, the minimum waist of the Gaussian beam is $\sim 6$ μm. The laser wavelength matches the surface plasmon resonance (SPR) peak of

the Au NPs in water. A high-speed camera captures the scattered light from the Au NPs to track their positions, and they are shown as glowing dots (Fig. 1b, c).

By observing the glowing dots (Supplementary Video 1), we find some fast NPs moving along the beam propagation direction ("positive motion") and some in the opposite direction ("negative motion"). NPs moving in both directions can have very high speeds, in stark contrast to the rest NPs experiencing Brownian motion. We single out several representative cases of ballistic NPs moving in either direction (see examples in Supplementary Video 2 for the positive motion and Supplementary Video 3 for the negative motion), track their positions as a function of time, and calculate their instantaneous speeds. We find that the maximum speed for the positive motion is 336,000 μm s$^{-1}$ and that for the negative motion is 245,000 μm s$^{-1}$ (see Fig. 1b, c). We also find that the average speed of the observed representative positive motions (204,000 μm s$^{-1}$) is higher than that of the negative motion (109,000 μm s$^{-1}$).

When the Gaussian beam illuminates the NP, there are mainly three types of possible forces, namely the optical force (radiative pressure)[17,19,20], the optical gradient force[17,19,20], and the photothermal gradient force[12,13,29]. Gravity and buoyancy are not considered because they are perpendicular to the optical axis. To identify the dominant force driving the ballistic NPs, we analyze several representative ballistic trajectories around the focal plane as shown in Fig. 1d, e. It is found that the NPs can cross the focal plane (black arrow in Fig. 1d, e) regardless of their moving directions. In addition, we can see that to the left (or right) side of the focal plane, ballistic NPs can move in both directions. These indicate that the ballistic NPs are not driven by any gradient (optical or photothermal) forces, since such forces are symmetric about the focal plane, which should converge the NPs from both sides of the focal plane towards it. We have also calculated the optical force field on the NP under the Gaussian beam used in the experiment (see Supplementary Note 1). We find that there is only positive force on the NP and the amplitude is almost symmetric to the focal plane. If gradient optical force is significant, it should have broken such a symmetry since it changes sign across the focal plane. We can thus infer that the gradient optical force is negligible compared to the dissipative optical forces. At lower NP concentrations, we also find ballistic NPs with both negative (Supplementary Video 4) and positive motions (Supplementary Video 5), suggesting that the ballistic movements should not be the result of the environment (e.g., scattered light from surrounding NPs or thermalization of water due to NP heating). Additionally, when we deposited the ballistic NPs on a quartz substrate and then characterized their geometrical configurations using a scanning electron microscope, it is found that the deposited NPs are isolated single NPs (see Supplementary Note 2). This also confirms that the ballistic movements are not related to the aggregation of NPs. These analyses leave the dissipative optical force (radiative pressure) as the only possible driver of the ballistic NPs.

To fully explain the observed ballistic NP movements, we need to understand how the NPs can move in speeds much higher than that of a typical optical force can sustain according to Stokes' law and why there are both optical pushing and pulling phenomena on some specific NPs. As discussed later, the optical force from our laser cannot sustain the observed high speed unless the effective viscosity experienced by the NPs is 100× lower than that of liquid water. The optical pulling is also impossible with our Gaussian beam irradiating an Au NP in a homogenous medium, unless there is a nearby dielectric structure optically coupled to the NP like the ones proposed in refs. [21,27].

According to several studies[30–33], when an Au NP is irradiated by a laser at the SPR peak, it can instantaneously generate a nano-

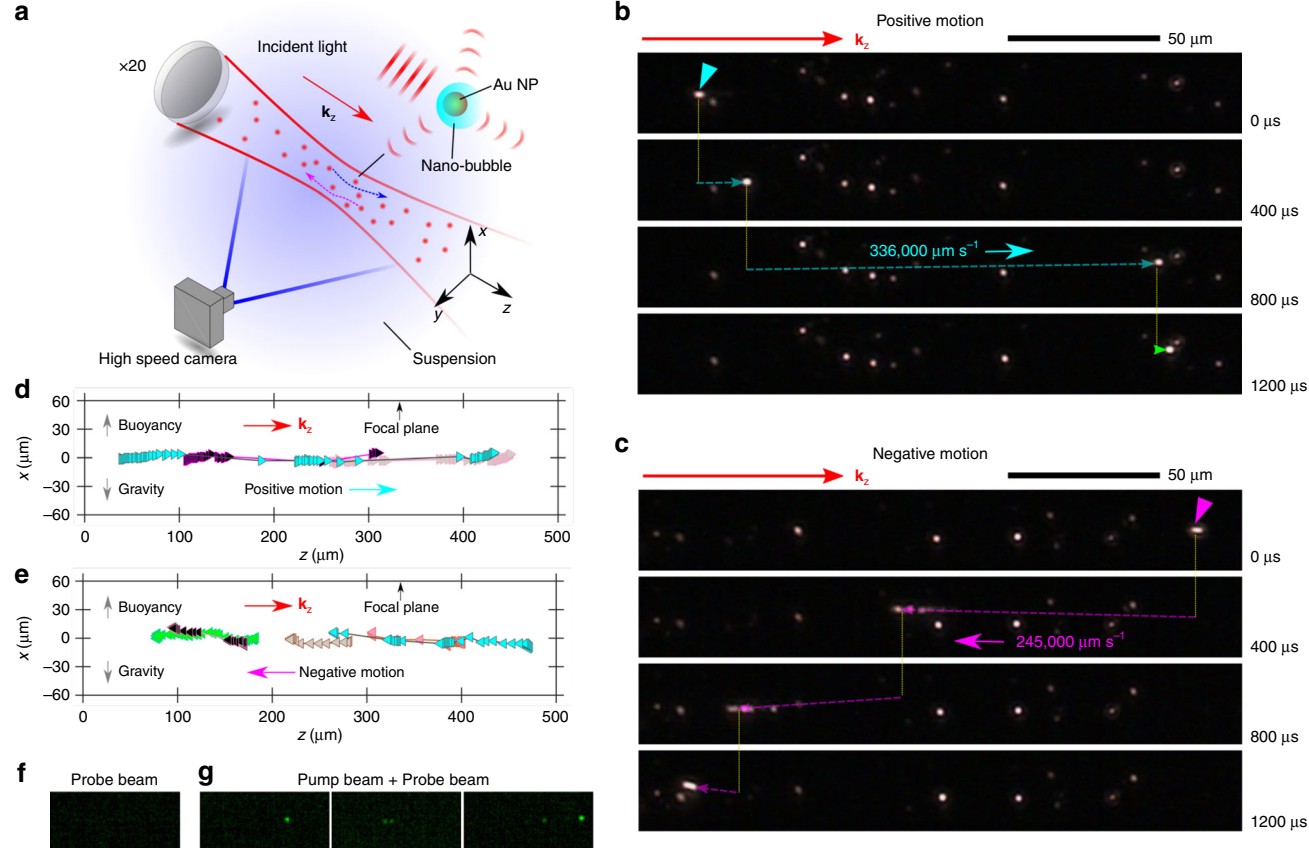

**Fig. 1 Observation of extremely fast ballistic movements of Au NPs. a** Schematic of the experimental system to monitor the dynamics of Au NPs dispersed in water. The red solid lines after the 20× lens indicate the Gaussian beam envelope. Au NPs scatter the incident light (red dots), which is captured by a high-speed camera with the field of view depicted by the blue solid lines. The blue and magenta arrows depict the Au NPs moving in the $+z$ and $-z$ directions, respectively referred to as the "positive motion" and the "negative motion". **b, c** Dark-field optical images (scale bars ~50 μm) of ultra-fast ballistic Au NPs with **b** the positive motion and **c** the negative motion. The cyan triangle in **b** and the magenta triangle in **c** indicate the positions of the selected Au NPs at $t = 0$. The cyan dotted line in **b** and the magenta dotted line in **c** are guide for the eye. The power of the laser is 690 mW, corresponding to 12 mW μm$^{-2}$ at the focal plane. **d, e** Representative **d** positive and **e** negative motion trajectories of the ballistic Au NPs projected onto the $x-z$ plane. Each color corresponds to a ballistic Au NPs. The time interval between two adjacent symbols is 400 μs. The black arrows at the top of the figures indicate the focal plane location. In **a-d**, $k_z$ is the wavevector of the incident laser light. **f, g** Images from the pump-probe optical scattering imaging experiment for **f** without pump beam and **g** with pump beam. Each green spot corresponds to the diffraction-limited scattered probe light from the Au NPs with nanobubbles.

bubble around it. Our heat transfer analysis also estimates that the plasmonic heating can rise the temperature of an NP inside a nanobubble to ~850 K (see Supplementary Note 3), which is much higher than the critical temperature of water (647 K). This calculated temperature from our simple model is also in reasonable agreement with that inferred from experiments (~1000 K)[34]. The hot NP thus can instantly evaporate water molecules, like the Leidenfrost effect, and can be fully encapsulated by vapor[30–33]. As long as the NP is illuminated by the laser, the NP can keep the high temperature. Using a pump-probe optical scattering imaging setup, we confirmed the existence of nanobubbles around the plasmonic Au NPs in our experiments when excited by the pump laser (Fig. 1f, g). This pump-probe method detects the intensity change of the scattered probe light (533 nm) at a certain solid angle due to the formation of nanobubble on Au NP under the illumination of the pump beam at the SPR peak (800 nm) (see Supplementary Note 4 for details). We note that the lifetimes of the plasmonic nanobubbles are reported to be ~200 ns (see refs. [30–33]). In our NP movement experiments, the time interval between laser pulses is ~12 ns, allowing the nanobubbles to shrink due to cooling by the

surrounding water[30–33,35,36], but they may not completely collapse since the pulse period (~12 ns) is shorter than the bubble lifetime (~200 ns). Dissolved gas in water can also contribute to the nanobubble volume, making it more stable than a pure vapor bubble[37].

**Optical forces on nanoparticle in nanobubble.** To understand how the nanobubbles may influence the optical force direction, we calculate the optical force in the beam propagation direction ($F_z$) on an NP with a nanobubble. We consider a bubble with a radius $r_{nb} < 120$ nm attached to the surface ($\theta$, $\varphi$) of an Au core–shell NP ($\theta$ and $\varphi$ are polar and azimuthal angles, Fig. 2a). When the bubble keeps growing, it will eventually encapsulate the NP, and we assume such a case for $r_{nb} > 120$ nm (i.e., inset in Fig. 2b). We have to note that the exact bubble radius for the attach-to-encapsulate transition will not influence the physics inferred from our analysis. Our assumption of the nanobubble nucleation and the encapsulation of NP is based on experimental studies, where Fu et al.[30] showed that upon laser excitation, a nanobubble can nucleate at a certain location on the Au NP

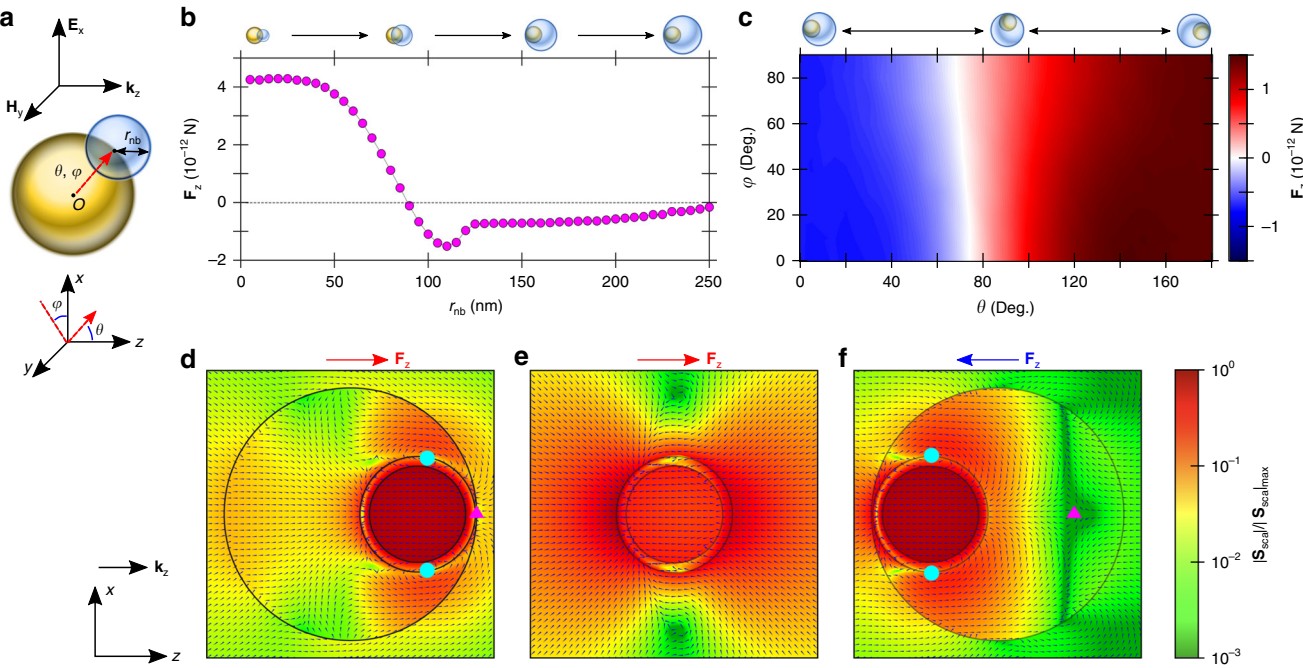

**Fig. 2 Optical force on an Au NP with a nanobubble. a** Schematic of an Au NP with a nanobubble. The Au NP (yellow sphere) consists of a 100 nm-diameter silica core and a 10 nm-thick Au shell. The nanobubble with a radius of $r_{nb}$ is attached to the surface ($\theta$, $\varphi$) of the Au NP, where $\theta$ and $\varphi$ are, respectively, the polar and the azimuthal angles in the polar coordinate with the origin ($o$) at the center of the Au NP. $\mathbf{E}_x$, $\mathbf{H}_y$, and $\mathbf{k}_z$ depict the electric field, the magnetic field, and the wavevector of the incident planewave, respectively. **b** The calculated optical force in the $z$-direction ($\mathbf{F}_z$) as a function of $r_{nb}$. Here, $\theta = 0^o$, $\varphi = 0^o$ and the amplitude of $\mathbf{E}_x$ is $2.6 \times 10^6$ V m$^{-1}$, corresponding to the laser in the experiment. The insets illustrate the schematic configurations of Au NP with a nanobubble with different $r_{nb}$. **c** The calculated $\mathbf{F}_z$ as a function of $\theta$ and $\varphi$. Here, $r_{nb} = 130$ nm and the amplitude of $\mathbf{E}_x$ is $2.6 \times 10^6$ V m$^{-1}$. On top of the contour, schematic configurations of Au NP with a nanobubble as a function of $\theta$ are illustrated. **d-f** The directions and normalized magnitudes of Poynting vectors of scattered fields $\mathbf{S}_{sca}$ for the structures of **d** $r_{nb} = 130$ nm and $\theta = 180^o$, **e** without a nanobubble, and **f** $r_{nb} = 130$ nm and $\theta = 0^o$. Directions of $\mathbf{F}_z$ are also shown.

surface and Lachaine et al.[32] showed that the nanobubble can fully encapsulate the NP if the fluence of the femtosecond laser pulse is higher than a threshold. For the NP used in this study, the threshold is around ~7 mJ cm$^{-2}$ (see ref. [32]), and our laser fluences are of 9–15 mJ cm$^{-2}$. The encapsulation of NP by the nanobubble is also intuitive given the very high temperature of the NP (>850 K), which is above the water critical temperature and can instantaneously evaporate liquid in contact with the hot NP surface. Since the beam waist (~6 μm) at the focal plane is much larger than the NP size (~120 nm), and the ballistic movements start at locations away from the focal plane, a linearly polarized planewave as an incident light is a good approximation[38] for estimating $\mathbf{F}_z$. In addition, since our laser pulses have an ultra-short duration (~94 fs), the time-averaged optical force by the pulsed laser can be approximated[39] by that from a continuous laser with the same central frequency and power density (see Supplementary Note 5). We calculate electromagnetic field distributions at various $r_{nb}$, $\theta$, and $\varphi$ using the finite element method (FEM) and then calculate $\mathbf{F}_z$ using the Maxwell stress tensor[20,27]. More details are provided in the "Method" section.

We find certain geometrical windows that induce either positive or negative $\mathbf{F}_z$ (Fig. 2b, c). For $r_{nb} > 90$ nm (Fig. 2b) and $\theta < 75^o$ (Fig. 2c), negative $\mathbf{F}_z$ can be achieved, where the nanobubble locates at the back side of the NP with respect to the incident light direction ($\mathbf{k}_z$). Otherwise, $\mathbf{F}_z$ is positive. $\mathbf{F}_z$ is found insensitive to the azimuthal angle $\varphi$ (Fig. 2c). While the positive $\mathbf{F}_z$ is intuitive, we need to understand the negative $\mathbf{F}_z$ on the NP under illumination of the single planewave. The optical force on the NP in the nanobubble can be decomposed

into two parts[40]: one from the interaction between scattered fields themselves ($\mathbf{F}_z^{ss}$), and the other from the interaction between the scattered field and the incident field ($\mathbf{F}_z^{si}$), and $\mathbf{F}_z = \mathbf{F}_z^{ss} + \mathbf{F}_z^{si}$. The sign and magnitude of each decomposed part can unveil which light field contributes dominantly to determining $\mathbf{F}_z$. For a representative case of negative motion ($r_{nb} = 130$ nm and $\theta = 0^o$), it is found that $\mathbf{F}_z^{ss}$ is responsible for the optical pulling force, where $\mathbf{F}_z^{ss}$ is $-1.254 \times 10^{-12}$ N, but $\mathbf{F}_z^{si}$ is only $4.99 \times 10^{-13}$ N. Since $\mathbf{F}_z^{ss}$ is due to the net momentum the NP can gain by compensating the net scattered photon momentum leaving the NP, if the scattering is isotropic, $\mathbf{F}_z^{ss}$ should be zero. Thus, the negative $\mathbf{F}_z^{ss}$ implies that there is uneven radiative scattering from the NP. We have investigated the energy flow pattern of scattered photons with the Poynting vector and found that at the backside of the NP, there is a "saddle" singular point, which induces unusual energy flows around the NP (see the magenta triangle in Fig. 2d). The saddle point turns the energy flow scattered from the NP to the back side of the NP and leads the energy to flow around the "vortex" singular points located at the sides of the NP (see the cyan circles in Fig. 2d). It is this circulation of the scattered energy that allows the NP to possess negative momentum. Our result corresponds well to an analysis introduced in ref. [41], where a particle irradiated by a Bessel beam can receive an optical pulling force when the pairs of saddle and vortex singular points, which are located at the backside of the NP, redirect and focus the energy flow into the particle. Without a nanobubble, the saddle point and the circulation of energy flow do not exist (see Fig. 2e), and the scattering is symmetric about the light polarization axis, leading to $\mathbf{F}_z^{ss} = 0$. This

indicates that the nanobubble is essential for forming the saddle singularity. We believe that the formation of a saddle point is due to the curvature of the nanobubble surface, as it works as an optical mirror to reflect the scattered light from the NP. We have also investigated the energy flow of scattered light for the representative case of positive motion ($r_{nb} = 130$ nm and $\theta = 180°$). As the NP is now placed close to the right nanobubble surface, the saddle point is sandwiched between the NP and the nanobubble (see Fig. 2f). While there are still two vortex points which can circulate the energy flow to be redirected into the NP, the strength of circulation is much suppressed in comparison to the negative motion case. As a result, the magnitude of optical pulling force that the NP receives is much weaker than the pushing force, leading to a positive $F_z^{ss}$ of $1.068 \times 10^{-12}$ N. We have also investigated the optical stress tensor profile on the surface of the Au NP for the representative cases (see Supplementary Note 6). Both cases show negative stress at the light-incoming side and positive stress at the rear side, and the sign of stress reverses as across the equator of the NP. For the negative motion case, it is observed that the negative stress around the side poles not only has a higher magnitude but also covers a larger area than those of the positive stress, thus leading to a net negative $F_z^{ss}$. We believe that the stronger negative stress is due to the strong energy circulation around the vortex singular point at the side poles of NP (see Fig. 2d). Meanwhile, for the positive motion case, the area and the magnitude of negative stress around the side poles are suppressed compared to that of the negative motion case, which yields a net positive $F_z^{ss}$. It is due to the

weakened energy circulation as the saddle point is squeezed between the surfaces of the NP and the bubble (see Fig. 2f). We have also calculated the far-field scattering patterns (Supplementary Note 7). It noted that the far-field scattering patterns do not intuitively correspond to the directions of optical forces on the NPs, which means that the optical force on the NP inside the nanobubble is the result of near-field scattering.

**Low dynamic viscosity experienced by ballistic nanoparticles**. Figure 3a shows the calculated $F_z$ on a supercavitating NP along the central axis of the Gaussian beam for two representative cases: one for the positive motion ($r_{nb} = 130$ nm and $\theta = 180°$) and one for the negative motion ($r_{nb} = 130$ nm and $\theta = 0°$). The NP is found to experience $F_z$ on the order of $\sim 10^{-12}$ N around the focal plane (Fig. 3a), with the negative $F_z$ uniformly smaller than the positive $F_z$ in amplitude. It is noted that the profiles of the optical forces, which correspond to the light intensity profile, confirm that the optical gradient forces are negligible, which would have changed the sign of the force across the focal plane. Meanwhile, the Stokes' friction force in a flow with a low Reynolds number (Re), which is Re = $\sim 10^{-3} \ll 1$ in our case, allows us to estimate the order of magnitude of the driving force ($F_{driving}$) on the ballistic NP given their measured speeds as: $F_{driving} = 6\pi\eta rv \approx 6\pi \times 8.9 \times 10^{-4}$ (kg m$^{-1}$ s$^{-1}$, $\eta$, viscosity of liquid water) $\times 6 \times 10^{-8}$ (m, $r$, radius of Au NP) $\times 1 \times 10^{-1}$ (m s$^{-1}$, $v$, speed of ballistic NP) = $\sim 10^{-10}$ N. This is two orders of magnitude larger than our calculated $|F_z|$ since we used liquid water viscosity. As the plasmonic Au NP is intensely excited by the laser to instantaneously evaporate the surrounding water and generate a nanobubble to encapsulate itself, i.e., supercavitation, the

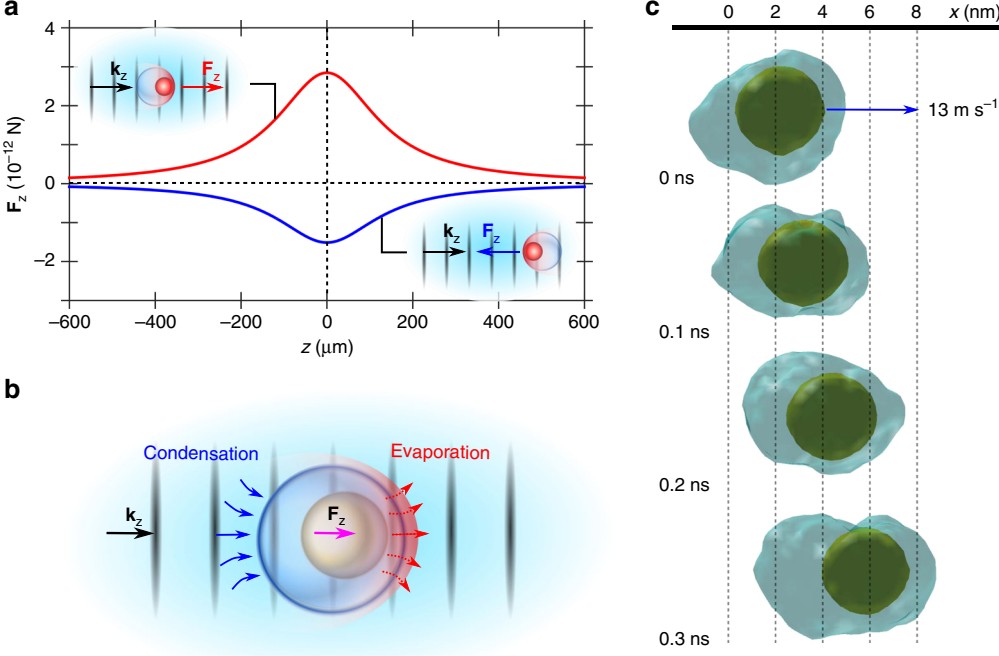

**Fig. 3 Positive and negative optical forces and the nanoscale Leidenfrost effect. a** The calculated $F_z$ on an Au NP with a nanobubble along the central axis (z-direction) of the Gaussian beam. A beam waist and intensity at the focal plane (z = 0) are 6 µm and 12 mW µm$^{-2}$, respectively. The red line represents the positive force for $r_{nb} = 130$ nm, $\theta = 180°$, and the blue line represents the negative force $r_{nb} = 130$ nm, $\theta = 0°$. The insets illustrate (left top) the positive motion and (right bottom) the negative motion. **b** Schematic illustration of a supercavitating ballistic Au NP. Under the laser illumination, the plasmonic Au NP is intensely heated to generate a nanobubble which encapsulates it. When the Au NP moves forward, it keeps evaporating water and maintaining a vapor cushion in front of it, which effectively makes the NP moving in a virtually frictionless environment, like the Leidenfrost effect. In the meantime, vapor at the trailing end of the bubble cools and condenses back to liquid as the hot NP moves forward. **c** Nanoscale Leidenfrost effect of a hot moving NP simulated by molecular dynamic (MD) simulations. The hot NP is thermostated at 1000 K (Supplementary Note 9 for justification), and the blue contour visualizes the isosurface of the critical density of liquid argon (0.536 g cm$^{-3}$), which represents the nanobubble surface. The NP moves in the positive x-direction with a speed of 13 m s$^{-1}$.

dynamic viscosity experienced by the NP should be different. Using the calculated optical force ($\sim 10^{-12}$ N) and Stokes' law, we can extract an effective dynamic viscosity ($\eta_{eff}$) experienced by the ballistic NPs as:

$$\eta_{eff} = \frac{(z_2 - z_1)}{6\pi r v} \left[ \int_{z_1}^{z2} \frac{1}{\mathbf{F}_z(z)} \mathrm{d}z \right]^{-1}, \qquad (1)$$

where $v$ is the experimentally determined average speed between positions $z_1$ and $z_2$ where the ballistic movement occurs. The averaged $\eta_{eff}$ (for the representative cases in Fig. 1) is $\sim 1.27 \times 10^{-5}$ kg m$^{-1}$ s$^{-1}$ for the negative motion and $\sim 1.03 \times 10^{-5}$ kg m$^{-1}$ s$^{-1}$ for the positive motion. These values are much smaller than the dynamic viscosity of liquid water but very close to that of vapor steam ($\sim 1 \times 10^{-5}$ kg m$^{-1}$ s$^{-1}$), suggesting that the ballistic NPs are moving in a gaseous environment. This can be interpreted as the following: as the laser-excited Au NP encapsulated by a bubble moves forward, it keeps evaporating water, maintaining a vapor cushion in front of it and extends the bubble boundary forward (Fig. 3b). This is very similar to the observed near-zero drag force on a hot metal sphere enclosed by a gas cavity due to the Leiden-frost effect[42] and in great analogy to the supercavitating torpedoes, which realized 5× faster speed than conventional ones[43]. As the NP moves, the trailing end of the bubble, which becomes further away from the hot NP, cools and vapor condenses back to liquid (Fig. 3b). This interpretation is well backed by the results of our molecular dynamics (MD) simulations. In Fig. 3c, we clearly see that a hot NP encapsulated by a vapor bubble can continuously evaporate liquid molecules in front of it as it moves forward with a speed of $\sim 13$ m s$^{-1}$, so that the bubble boundary is extended and the NP is always enclosed in a gaseous environment (see Supplementary Notes 8 and 9 for simulation details). While this simulation uses a Lennard-Jones argon model system, the physics should be the same and analogy can be drawn with our NP-in-water experiment. We note that the 13 m s$^{-1}$ speed simulated is much higher than our experimental observation (0.1–0.3 m s$^{-1}$), but the NP is still found able to instantaneously extend the bubble front while moving. We also note that the bubble can experience optical forces as well, but they are usually positive with amplitude calculated to be around $6.0 \times 10^{-13}$ N. However, with such forces, the nanobubble can only move with a speed more than two orders of magnitude slower (e.g., Stokes's law yields a speed of $\sim 410$ μm s$^{-1}$) than the NPs. Therefore, it should be the instantaneous evaporation that keeps the NP inside a gaseous environment, instead of the synchronization between the dynamical motions of the NP and the bubble. We mention that there is also electrostriction[44] forces that can deform the nanobubble, but in our scenario it should not be important as new interfaces of nanobubble is formed by the hot moving NP.

## Discussion

Since nanobubble dynamics is stochastic, not all Au NPs are enclosed by nanobubbles to exhibit the ballistic movement. We can see in Supplementary Video 1 that the ballistic motion generally occurs within 300 μm at either side of the focal plane with a significant portion of the NPs in this region exhibiting ballistic motion. This is because the laser intensity is sufficiently high in this region. NPs outside of this region are mainly seen drifting due to thermal convection perpendicular to the laser beam direction. However, even within this region, not every NP is guaranteed to move ballistically since bubble generation is inherently statistical and the probability may depend on factors like defects on the NP surface and geometry of the NP (e.g., defects and sharp corners are known to change bubble nucleation threshold[45]). If a nanobubble is generated and the NP is encapsulated, moving forward or backward still would depend on the relative position of the NP

inside the bubble (as discussed in Fig. 2). In addition, Fig. 2b, c show that the geometrical window for the negative $\mathbf{F}_z$ is narrower than that for the positive $\mathbf{F}_z$, which may explain our observation that more ballistic NPs move positively than negatively (Supplementary Video 1). The magnitude of the positive $\mathbf{F}_z$ being generally larger than the negative $\mathbf{F}_z$ (Fig. 3a) also explains the observed higher average speed of the positive motion than that of the negative motion (Fig. 1). Although $\mathbf{F}_z$ depends on the instantaneous geometrical configuration of the NP-bubble structure and thus the forces should be dynamic, we observe that the NPs move almost strictly along the beam axis (Fig. 1d, e). Moving in a virtually frictionless supercavitation, the ballistic NPs exhibit speed-to-body length ratios greater than $10^6$. This is three to seven orders of magnitude higher than other directed nanoswimmers/microswimmers and one to five orders of magnitude higher than random microswimmers (Fig. 4).

In theory, any nanoscale objects that can have supercavitation after laser excitation should exhibit similar ballistic movements. To confirm that this effect is generalizable, we have performed experiments with Au nanorods (NRs), which also have a SPR peak of 800 nm. It is observed that Au NRs also exhibit similar ballistic movements with an observed speed of $\sim 13,000$ μm s$^{-1}$ (Supplementary Note 10 and Supplementary Video 6). Since the SPR characteristic of Au NR is anisotropic due to its shape, only the NRs that align with the incident electric field direction can experience strong SPR coupling and lead to supercavitation. Thus, the ballistic movements of Au NRs may easily be interrupted, because the different forces (e.g., optical force and drag force) can make NRs to misalign with the electric field. However, in Supplementary Note 10, we can clearly see a glowing dot moving for $\sim 13$ μm within 1 ms along the beam propagating direction

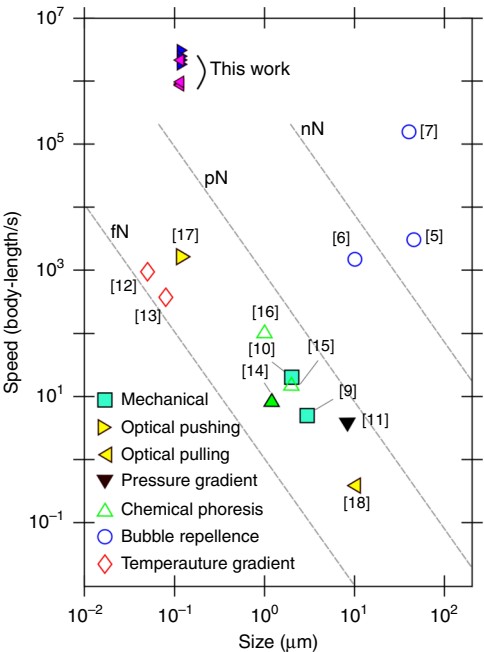

**Fig. 4 Comparison of the ballistic Au NP swimmers with other swimmers.** Top-four highest speeds (measured in body-length s$^{-1}$) for the negative and the positive motion compared to those of the nanoswimmers/microswimmers in the literature. The filled symbols represent nanoswimmers/microswimmers whose movement can be directed. The hollow symbols represent those moving in random directions. The legend indicates propulsion mechanisms. The gray dash lines are contour lines of the magnitude of the force needed to achieve the corresponding speed of a swimmer in liquid water.

(see also Supplementary Video 6). This ballistic Au NR shows a peak speed of 260,000 body-length $s^{-1}$, which is still ~$10^2$ to $10^5$ times faster than the reported nanoswimmers/microswimmers driven by optical forces[17,18]. With the calculated optical force ($1 \times 10^{-13}$ N–$7 \times 10^{-13}$ N) on the Au NR (see Supplementary Note 10), Stokes' law implies a dynamic viscosity of $4.4 \times 10^{-5}$ kg $m^{-1}s^{-1}$, which is much lower than that of liquid water and on the same order of magnitude as that of vapor. We have not observed any negative motion for Au nanorods, most likely due to the stringent alignment requirements that limit the probability of enabling supercavitation and negative optical forces. Nevertheless, this result demonstrates that the ballistic movement can be potentially generalized to other kinds of plasmonic NPs with different composition, geometry and dimensions. We believe that with proper NP design, both the ballistic motion and optical force direction can be manipulated and optimized.

## Methods

**Sample preparation**. A quartz cuvette (Hellma, Sigma-Aldrich, 10 mm × 10 mm) with four windows is pre-cleaned with acetone, isopropyl alcohol and deionized water in an ultrasonic bath. A silica-Au core–shell NP-water suspension (Auroshell, Nanospectra Biosciences, Inc) with a NP number density of $2 \times 10^{15}$ m$^{-3}$ or an Au nanorod NP-water suspension (NanoXact, nanoComposix) with a NP number density of $2.6 \times 10^{17}$ m$^{-3}$ is mixed with deionized water to achieve different number densities. The NP number densities are: Fig. 1b–e and Supplementary Videos 1–3 (for the core–shell NP): $4 \times 10^{14}$ m$^{-3}$, Supplementary Video 4 (for the core-shell NP): $5 \times 10^{13}$ m$^{-3}$, Supplementary Video 5 (for the core–shell NP): $1 \times 10^{13}$ m$^{-3}$, and Supplementary Video 6 (for the Au NR): $1.3 \times 10^{17}$ m$^{-3}$. The cleaned cuvettes filled with these NP-water suspensions are used for the experiments.

**Ultrafast laser experiment and characterization**. Mode-locked femtosecond laser pulses (center wavelength of 800 nm and a full-width half maximum of 10 nm) are emitted from a Ti:sapphire crystal in an optical cavity (Spectra Physics, Tsunami). The laser is collimated and a linearly polarized TEM$_{00}$ mode beam. The time interval between pulses is 12.4 ns (80.7 MHz), and the pulse duration is ~94 fs. The power of the laser is fixed to 690 mW. The femtosecond laser passes through a 20× objective lens and focuses on the NP-water suspension in the cuvette. A beam profiler (Thorlab BP104-UV) measures the minimum waist of the Gaussian beam (~6 μm). We note ballistic movements were still observed with a 10× objective lens (Supplementary Video 2) or a laser power as high as ~1 W. The scattered lights from the Au NPs passes through another objective lens and focuses on the image sensors of the high-speed camera (HX-7, NAC). The recorded images in the camera are then analyzed using a customized image processing software in MATLAB.

**Calculation of optical force**. The FEM software (RF module, COMSOL Multiphysics) is used to calculate the optical forces on different Au NPs with nanobubbles (structures illustrated in Fig. 2a). The Au NP consists of a silica core (radius of 50 nm) and an Au shell (thickness of 10 nm). It is modeled that an Au NP with a nanobubble is immersed in an infinite volume of water. In the simulations, for the infinite volume of water, a spherical volume of water (radius of 400 nm) is enclosed by a perfectly matched laser with a thickness of 130 nm. The Au NP with a nanobubble is located at the center of the water sphere. The refractive indices of water, air, and silica at the wavelength of 800 nm are 1.33, 1.00, and 1.50, respectively. The real and complex parts of the refractive index of the Au shell at the wavelength of 800 nm are 0.154 and 4.91, respectively. A linearly polarized planewave is modeled as the incident light for Fig. 2b–f. A Gaussian beam given by the paraxial approximation[38] is used for Fig. 3a and extracting $\eta_{eff}$, where it has a minimum waist of 6 μm and the maximum light intensity of 12 mW μm$^{-2}$, corresponding to the experimental conditions. After solving the electromagnetic field distribution of the simulation domain, optical force (**F**) is calculated as:

$$\mathbf{F} = \oiint T \cdot \mathbf{n}\, \mathrm{d}a,$$

where $T$ is the time-averaged Maxwell stress tensor[20,27], and **n** is the normal vector of the surface of the Au NP. We use the time-averaged Maxwell stress tensor to calculate **F** in the $z$-direction on the surface of the Au NP.

## Data availability

All datasets generated during and/or analyzed during the current study are available from the corresponding author on reasonable request.

## Code availability

The code used for data analysis as well as for display of the data is available from the corresponding author upon reasonable request.

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

## Acknowledgements

This work is supported by National Science Foundation (1706039 and 1937923) and the Center for the Advancement of Science in Space (GA-2018-268).

## Author contributions

E.L. and T.L. designed the experiments. E.L. set up and performed the experiments, and performed the optical simulations. D.H. and T.L. performed the MD simulations. All authors discussed the results and the mechanism of the ballistic Au NPs. E.L. and T.L. wrote the manuscript.

## Competing interests

The authors declare no competing interests.
