## [Peer Review File · Nature Communications]

Editorial Note: Parts of this Peer Review File have been redacted as indicated where no third party permissions could be obtained.

Reviewers' comments:

Reviewer #1 (Remarks to the Author):

The authors report a study of optically induced forces on metallic nanoparticles suspended in an aqueous environment. The optical irradiation is in conditions of surface plasmon resonance. The authors observe that the nanoparticles moves with high velocities, which they call "unprecedented". I would recommend using a different terminology instead of swimmers for those 60nm metallic particles exposed to a laser beam and also a different velocity unity instead of body-length per second when, in fact, one can't even resolve the size of the object that moves in front of the camera!

Nevertheless, the reported experiments were carefully conducted. An independent pump-probe experiment suggests that gas bubbles are sometimes formed around the metallic particles and it is argued that this is the reason for the high velocities recorded. This is an intriguing hypothesis and, to support it and to understand some of its implications, the authors perform calculations of the optical force. Unfortunately, this part is rather sketchy.

First of all, it is not clear how the Maxwell stress tensor is calculated. There two types of interfaces here: AU-gas and gas-liquid. Which ones are included? I understand that the results presented refer to the force acting on NP but how about the action on the bubble itself? In this respect, the authors can consult, for instance, J. Opt. Soc. Am. B 30(6), 1694 (2013).

The second aspect is the type of excitation. It is know that when radiation is pulsed, the interaction is modified especially if the object supports surface states. See for instance Phys. Rev. Lett. 102, 050403 (2009). This leads to significant differences between how the momentum is transferred to matter and raises even fundamental issues related to Minkowski and Abraham interpretations, see Phys. Rev. E 73, 056604 (2006). About this aspect one can also consult Phys Rev E 79, 026608 (2009). A full description that accounts for both the temporal and the spatial (core-shell) asymmetries of the interaction could also explain the direction of the overall force.

Finally, does their description of the force affect the interpretation of the dynamic viscosity in terms of supercavitation? A more rigorous electromagnetic treatment of this interesting physical situation is required. A more detailed numerical estimation of the optically induced force should permit a direct comparison with the forced inferred from the experimental observations. I find that the lack of a direct comparison between the model's outcome and experiment is the main deficiency of this manuscript. I think it can be corrected by a more realistic modeling of the electromagnetic interaction. Otherwise, one can only claim the observation of an intriguing consequence of focusing a femtosecond laser into a colloidal suspension of metallic nanoparticles.

For these reasons, I cannot recommend publication of this manuscript.

Reviewer #2 (Remarks to the Author):

Optical manipulation of nanoparticles has been extensively investigated in recent years in many physics, such as molecular machinery, nanorobotics, and drug delivery. Especially, negative optical force is an interesting field in light-mater interaction. In this paper, the authors report optical pushing and pulling force of plasmonic nanoparticles with super-fast speeds. The numerical simulation and experimental results are structured well, which can fully support their claims. I think this article can be considered after addressing the following issues.

(I) The authors give the conditions (geometrical windows) for obtaining negative optical force. And they explain it with negative light direction (k_z). But what are the physical reasons for this counter-intuitive phenomenon? The authors could plot the energy flow around the nanoparticles.

The singular points of Poynting vector (see Laser Photon. Rev. 9, 75(2015)) may play similar roles as the nano-bubble at the back side of the nanoparticles in your work.

(II) What is the force density distribution around the nanoparticles for the pulling and pushing case? And are there any significant difference of far-field scattering for these two cases? These may also help to understand the direction of the optical force.

In summary, I would recommend it for publication if my concerns are sufficiently considered.

Reviewer #3 (Remarks to the Author):

The authors reported the design and optically-driven motions of ultrafast nanoswimmers via a new mechanism of bubble wrapping. There are certainly innovative and exciting aspects of the work. However, the paper needs to be significantly revised to make sure some of their conclusions and terminology are correctly described, especially the key mechanism of optical supercavitation that the authors argued. Detailed suggestions are as below:

1. For their discussions on Figure 1b and 1c, the ultrafast motions of the nanoswimmers: their measurement of the speed is based on movies like Supplementary Video 1. However, Supplementary Video 1 as it is now has both particles that are moving superfast and those are only moving locally. Can the authors clearly single out/label the particles based on which they argue about the superfast motions, and detail how the average speed was calculated (instantaneous or averaged speed?). Also explain why there are heterogeneity in the particle motions – is that due to synthetic heterogeneity of the particles themselves or just because the light intensity is uneven? There are some discussions about this on Page 8, which don't alleviate the concern that the light pulse is not a truly robust and reproducible method to drive the particles. Moreover, the particle concentration in Supplementary Video 1 is very high, which can make it difficult to track single particle motions (i.e., how to link the particles across frames as the same particle). Do the authors have low particle concentration movies, without ambiguities on tracking?
2. The discussions on the three possible optical forces on page 5 in the main text: can the authors make them more quantitative? How small is "small"? Currently the discussions are very qualitative. One can only rule out these effects after quantitatively comparing them with the supercavitation effects.
3. On the optical force calculation presented on page 6: It is not clear why a nanobubble of a size larger than the NP suggests complete enveloping; the nanoparticles can have a big nanobubble attached only partially on the NP surface, which is the mechanism for numerous literatures on self-propelled micron-sized particles. It is not clear why the bubble would want to completely wet the particle surface and create this frictionless environment.
4. How does photothermal effect play a role now that the light is at the plasmonic resonance wavelength of the gold NP?
5. In their last discussion on gold nanorods, again the explanation of the direction dependence and the lack of negative motions is very hand-wavy. The authors need to articulate in a logically way, with supplementary notes as appropriate, using experimental data to verify their discussions, instead of just making unverified statements. For example, for the alignment argument: can the authors show what is the orientation of gold nanorods as they exhibit positive motions? Parallel or perpendicular to the motion direction?

We have carefully studied the reviewers' comments and included our point-by-point responses below. We have also modified the manuscript accordingly, and the changes are highlighted.

Reviewer #1 (Remarks to the Author):

Comment 1. The authors report a study of optically induced forces on metallic nanoparticles suspended in an aqueous environment. The optical irradiation is in conditions of surface plasmon resonance. The authors observe that the nanoparticles move with high velocities, which they call “unprecedented”. I would recommend using a different terminology instead of swimmers for those 60nm metallic particles exposed to a laser beam and also a different velocity unit instead of body-length per second when, in fact, one can't even resolve the size of the object that moves in front of the camera!

Response: We thank the reviewer for these comments. We agree that it is difficult to resolve the physical size of a nanoparticle in front of the camera. But each NP has a well-defined size and as long as they are not aggregated, we know the size of the moving particle precisely. To address this issue, we have deposited the ballistic NPs on a quartz substrate using the optical force and then characterized their size using scanning electron microscope (SEM). We have leveraged the uniqueness of the negative motion of NPs to avoid the deposition of non-supercavitating NPs, as the optical pulling force on an NP is only made possible with supercavitation. As the SEM image in **Fig. S2** shows, the NPs deposited on the quartz surface are isolated single NPs and their sizes are well-defined (diameter of 120 nm).

We have addressed this concern in the revised manuscript as the following:

On page 5, at lines 19 ~ 22:

“Additionally, when we deposited the ballistic NPs on a quartz substrate and then characterized their geometrical configurations using a scanning electron microscope, it is found that the deposited NPs are isolated single NPs (see Supplementary Materials Section SM2). This also confirms that the ballistic movements are not related to the aggregation of NPs.”

, and in Supplementary Material Section SM2:

“Section SM2. Deposited Ballistic NPs on Quartz Substrates

We have leveraged the uniqueness of the negative motion of NPs to avoid the deposition of non-supercavitating NPs, as the optical pulling force on an NP is only made possible with supercavitation. As the SEM image in **Fig. S2b** shows, the NPs on the surface of the quartz are single NPs and their sizes are well-defined.

Figure S2 | (a) An experimental setup to deposit the ballistic NPs with negative motion on a quartz substrate. An objective lens (10×, beam waist $\sim 20 \mu\text{m}$) was used to focus the femtosecond laser with a power of 690 mW on the substrate surface. (b) The collected ballistic NPs on the quartz substrate visualized using SEM.”.

In addition, in revised manuscript, the title has been changed to “**Ballistic Supercavitating Nanoparticles Driven by Single Gaussian Beam Optical Pushing and Pulling forces**” to clearly indicate that the moving objects are NPs.

However, we would like to keep the unit of body-length per second because of the following two reasons: 1) In many references, moving NPs driven by external energy sources or by self-propelled mechanisms have been considered as “nanoswimmers”^{1,2}, and 2) in the literature of micro/nanoswimmers, it is a common practice to use the unit of body-length per second to fairly compare the speed of swimmers with difference sizes.

Comment 2. Nevertheless, the reported experiments were carefully conducted. An independent pump-probe experiment suggests that gas bubbles are sometimes formed around the metallic particles and it is argued that this is the reason for the high velocities recorded. This is an intriguing hypothesis and, to support it and to understand some of its implications, the authors perform calculations of the optical force. Unfortunately, this part is rather sketchy.

First of all, it is not clear how the Maxwell stress tensor is calculated. There two types of interfaces

here: AU-gas and gas-liquid. Which ones are included? I understand that the results presented refer to the force acting on NP but how about the action on the bubble itself? In this respect, the authors can consult, for instance, J. Opt. Soc. Am. B 30(6), 1694 (2013).

Response: We thank the reviewer for these questions. Yes, the optical force was calculated by integrating the Maxwell stress tensor over the Au NP/vapor interfaces. So, it was the force acting on the Au NP. Before discussing the optical force on the nanobubble, we would like to first show the rise of temperature of the NP due to the plasmonic heating from the laser. We have calculated the time-averaged optical absorptions and estimated how the absorbed energy can heat up the NP using the finite element method to solve the continuity, momentum and heat transfer equations of the NP-in-nanobubble system in water. The temperature profile of the system is shown in **Fig. S3 in the revised Supplementary Materials**, where the NP temperature can reach ~ 850 K, much higher than the critical temperature of water (647 K). This calculated temperature from our simple heat transfer model is also not far from that inferred from experiments (~ 1000 K)³. At such a high temperature, water molecules come in contact with the hot NP can be instantly vaporized into the gaseous phase – like the Leidenfrost effect. While the focus of this study is not on the mechanism of nanobubble formation, there have been a number of reports in the literature showing that nanobubble can be generated instantly by laser irradiation at the NP SPR^{4,5} as we have cited in the manuscript. In addition, we have ultra-fast pulsed laser with a repetition rate of 80.7 MHz. Our calculation has shown that the temperature would only decrease by a few degrees before it is heated by another pulse (see calculation in the original Supplementary Materials). It is thus reasonable to say that wherever the hot NP is moved to by the optical force, it can instantly evaporate water molecules. Thus, a new interface of nanobubble at the front of the NP can be developed wherever the hot NP moves to, i.e., it is not the same bubble moving together with the NP, but the boundary of the bubble is extended by instantaneous evaporation. This was also confirmed by our MD simulations in the paper.

We have addressed this concern in the revised manuscript as the following:

On page 6, in lines 5 ~ 11:

“Our heat transfer analysis also estimates that the plasmonic heating can rise the temperature of an NP inside a nanobubble to ~ 850 K (see Supplementary Material Section SM3), which is much higher than the critical temperature of water (647 K). This calculated temperature from our simple model is also in reasonable agreement with that inferred from experiments (~ 1000 K)³⁴. The hot NP thus can instantly evaporate water molecules, like the Leidenfrost effect, and can be fully encapsulated by vapor³⁰⁻³³. As long as the NP is illuminated by the laser, the NP can keep the high temperature.”

, in Reference:

- “30. Fu, X., Chen, B., Tang, J. & Zewail, A. H. Photoinduced nanobubble-driven superfast diffusion of nanoparticles imaged by 4D electron microscopy. *Sci. Adv.* **3**, e1701160 (2017).
31. Lukianova-Hleb, E., Volkov, A. N. & Lapotko, D. O. Laser pulse duration is critical for generation of plasmonic nanobubbles. *Langmuir* **30**, 7425-7434 (2014).
32. Lachaine, R. et al. Rational design of plasmonic nanoparticles for enhanced cavitation and cell perforation. *Nano Lett.* **16**, 3187-3194 (2016).
33. Metwally, K., Mensah, S. & Baffou, G. Fluence threshold for photothermal bubble generation using plasmonic nanoparticles. *J. Phys. Chem. C* **119**, 28586-28596 (2015).
34. Hodak, J. H., Henglein, A., Giersig, M. & Hartland, G. V. Laser-induced inter-diffusion in AuAg Core-Shell Nanoparticles. *J. Phys. Chem. B* **104**, 11708-11718 (2000).”

, and in Supplementary Material Section SM3:

“**Section SM3. Temperature of Plasmonic Au NP in a Nanobubble**

Figure S3| The calculated temperature of the NP-in-nanobubble system when the amplitude of the electric field of an incident planewave is 2.6×10^6 V/m, corresponding to the experimental laser condition.”.

In the meantime, the optical forces on the nanobubble, given by the time-averaged Maxwell stress tensor, are usually positive with magnitudes comparable to those on the NPs. The calculated values are around 6.0×10^{-13} N with slight variations depending on where the NP is inside the bubble. With this magnitude of force, a nanobubble itself can only move with a speed more than two orders of magnitude slower (Stokes's law yields a speed of $\sim 410 \mu\text{m s}^{-1}$) than the ballistic NPs, as the nanobubble is dragged by the viscosity of liquid water. However, as discussed above, the movement of the bubble is not very important as it is the instantaneous evaporation that keeps the NP inside a bubble, instead of the synchronization between the dynamical motions of the NP and the bubble. The paper referenced by the reviewer also mentioned electrostriction forces. It is noted that the deformation of nanobubble due to the electrostriction force should not be important as a new interface of nanobubble is formed by the hot moving NP.

We have addressed this concern in the revised manuscript as the following:

From page 10, line 7 ~ 15:

“We also note that the bubble can experience optical forces as well, but they are usually positive with amplitude calculated to be around 6.0×10^{-13} N. However, with such forces, the nanobubble can only move with a speed more than two orders of magnitude slower (e.g., Stokes's law yields a speed of $\sim 410 \mu\text{m s}^{-1}$) than the NPs. Therefore, it should be the instantaneous evaporation that keeps the NP inside a gaseous environment, instead of the synchronization between the dynamical motions of the NP and the bubble. We mention that there is also electrostriction⁴⁴ forces that can deform the nanobubble, but in our scenario it should not be important as new interfaces of nanobubble is formed by the hot moving NP.”

, and in Reference:

“44. Ellingsen, S. A. Theory of microdroplet and microbubble deformation by Gaussian laser beam. *J. Opt. Soc. Am. B.* **30**, 1694-1710 (2013).”

Comment 3. The second aspect is the type of excitation. It is know that when radiation is pulsed, the interaction is modified especially if the object supports surface states. See for instance Phys. Rev. Lett. 102, 050403 (2009). This leads to significant differences between how the momentum is transferred to matter and raises even fundamental issues related to Minkowski and Abraham interpretations, see Phys. Rev. E 73, 056604 (2006). About this aspect one can also consult Phys Rev E 79, 026608 (2009). A full description that accounts for both the temporal and the spatial (core-shell) asymmetries of the interaction could also explain the direction of the overall force.

Response: We thank the reviewer for this comment. To address this issue, we carefully read the papers suggested by the reviewer, and also studied the optical force by pulsed laser. We agree that

the optical force by a pulsed laser can be different from the force by a continuous wave laser as a function of time. But, our analysis ensures that the time-averaged optical force by the pulsed light can be approximated by the optical force from a continuous wave light with the same central frequency and power density.

We have addressed this concern in the revised manuscript as the following:

On page 7, in lines 8 ~ 11:

“In addition, since our laser pulses have an ultra-short duration (~94 fs), the time-averaged optical force by the pulsed laser can be approximated³⁹ by that from a continuous laser with the same central frequency and power density (see Supplementary Material Section SM5).”

, in Reference:

“39. Preez-Wilkinson, N. et al. Forces due to pulsed beams in optical tweezers: linear effects. *Opt. Express* **23**, 7190-7208 (2015).”

, and in Supplementary Material Section SM5:

“Section SM5. Approximation of the Time-averaged Optical Force from the Pulsed Laser with the Force from a Continuous Single Plane Wave

The optical force by a pulsed laser can be a function of time, and it can be described by Lorentz’s force (f) density equation on an object:

$$f - \frac{\partial S}{\partial t} = \nabla \cdot T \quad (S1)$$

where T is the Maxwell stress tensor, and S is the electromagnetic momentum density (e.g., $D \times B$ by Minkowski momentum or $(E \times H)/c^2$ by Abraham momentum)^{6–8}. It is clear that f can be a function of time (t) if the incident light is a function of time. When the incident light is a pulsed beam, the amplitude oscillation of the electromagnetic field at the optical frequency is convoluted by the duration of a pulse. In our case, the duration is ~ 94 fs, and the optical frequency is 3.7×10^{14} Hz (= a period of ~2.7 fs). These time scales are much faster than the mechanical response of the ballistic NPs. For example, at ~ 94 fs, a ballistic NP with a velocity of 100,000 $\mu\text{m/s}$ can only move $\sim O(10^{-14})$ m. This suggests that averaging the force in time should be appropriate for analyzing the motion of NPs. These are well described in elsewhere^{9–11}. Using Eq. S1, the time-averaged force $\langle F \rangle_t$ on an object can be written as:

$$\langle F \rangle_t = \frac{1}{t_2 - t_1} \left\{ \int_{t_1}^{t_2} \oint_A T \cdot dA dt - \int_V S(t = t_2) dV + \int_V S(t = t_1) dV \right\} \quad (\text{S2})$$

where A and V are the surface and the volume of the object, respectively. Since the pulsed laser has a repetition rate of $\nu_0 = 80.7$ MHz, if we pick t_1 and t_2 to integrate over one pulse, the last two volume integral terms in Eq. S2 vanish as they are identical. It means that the momentum density terms are not needed from the perspective of the time-averaged force, and the Maxwell stress tensor alone is sufficient to determine $\langle F \rangle_t$. Then, we check if $\langle F \rangle_t$ from a pulsed laser is the same as that from a continuous laser with the same power density. To check this, we define the pulsed incident light as the summation of dispersive planewaves, where each planewave is linearly polarized in the x -direction and propagates in the z -direction:

$$E(t, z) = \hat{x} \frac{1}{\nu_0} \int_{-\infty}^{+\infty} E_\nu(\nu) e^{i(2\pi\nu)(t - \frac{n_m z}{c_0})} d\nu \quad (\text{S3})$$

where $E(z, t)$ is the electric field as a function of space and time, c_0 is the speed of light in vacuum, and n_m is the refractive index of the medium. $E_\nu(\nu)$ is the amplitude of electric field as a function of frequency (ν) and can be expressed as a Gaussian function:

$$E_\nu(\nu) = E_0 \sqrt{\nu_0} \left(\frac{2\pi}{a} \right)^{\frac{1}{4}} e^{-\left(\frac{\pi^2 (\nu - \frac{\omega_c}{2\pi})^2}{a} \right)} \quad (\text{S4})$$

where ω_c is the central mode angular frequency of the pulsed light (2.35×10^{15} Hz, corresponding to a vacuum wavelength of 800 nm), $a = 2\ln 2/\tau^2$, and E_0 is a coefficient in the unit of V/m, which gives the time-averaged power flux of a pulse as $\frac{1}{2} c_0 \epsilon_0 n_m E_0^2$. Since the duration of the laser in our experiment is ~ 94 fs, which is an ultra-short pulse, the time-averaged force $\langle F \rangle_t$ can be approximated as the summation of the forces from every planewave components in the pulsed light. Subsequently, Eq. S2 can be re-written with Eq. S4 as:

$$\langle F \rangle_t = \frac{1}{t_2 - t_1} \int_{t_1}^{t_2} \oint_A T \cdot dA dt \approx \int_{-\infty}^{+\infty} \frac{|E_\nu(\nu)|^2}{\nu_0} F(\nu) d\nu \quad (\text{S5})$$

where $F(\nu)$ is the optical force when a single planewave at the frequency of ν with the unity amplitude of electric field is incident to the object, and it can be calculated by $\oint_A \langle T \rangle_t \cdot dA$; $\langle T \rangle_t$ is the time-averaged Maxwell stress tensor given by the single planewave; $F(\nu)$ is in the unit of $\text{N}/(\text{V/m})^2$. Thus, we can estimate $\langle F \rangle_t$ by calculating $F(\nu)$ in the frequency domain. Fig. S6 shows

$|E_\nu(\nu)|^2/\nu_0$ and $F(\nu)$ as a function of ν , where E_0 is 2.6×10^6 V/m corresponding to the light intensity in our experiment. To study the positive and negative motions of ballistic NPs, representative cases are chosen to calculate $F(\nu)$. These include $r_{nb} = 130$ nm and $\theta = 180^\circ$ and $r_{nb} = 130$ nm and $\theta = 0^\circ$ (refer to Fig. 2 in main text for coordinates). Eq. S5 gives $\langle F \rangle_t$ as 1.413×10^{12} N for the case of $r_{nb} = 130$ nm and $\theta = 180^\circ$ and -7.553×10^{13} N for the case of $r_{nb} = 130$ nm and $\theta = 0^\circ$. These are almost identical to the optical forces given by the continuous planewave at a central frequency of $\frac{\omega_c}{2\pi} = 3.7474 \times 10^{14}$ Hz (or $\lambda = 800$ nm), which are 1.411×10^{12} N for the case of $r_{nb} = 130$ nm and $\theta = 180^\circ$ and -7.554×10^{13} N for the case of $r_{nb} = 130$ nm and $\theta = 0^\circ$. These results ensure that the time-averaged optical force by the pulsed light can be approximated as the optical force from a continuous wave light with the same central frequency and power density.

Figure S6 | The calculated (a) $|E_\nu(\nu)|^2/\nu_0$ and (b) $F(\nu)$ as a function of ν when E_0 is 2.6×10^6 V/m, corresponding to our laser in the experiment.”.

Comment 4. Finally, does their description of the force affect the interpretation of the dynamic viscosity in terms of supercavitation? A more rigorous electromagnetic treatment of this interesting physical situation is required. A more detailed numerical estimation of the optically induced force should permit a direct comparison with the forced inferred from the experimental observations. I find that the lack of a direct comparison between the model’s outcome and experiment is the main deficiency of this manuscript. I think it can be corrected by a more realistic modeling of the electromagnetic interaction. Otherwise, one can only claim the observation of an intriguing consequence of focusing a femtosecond laser into a colloidal suspension of metallic nanoparticles.

Response: We thank the reviewer for these questions. We have now performed more realistic

optical calculations with the pulsed laser, as described in our response to **Comment 3**. The analysis shows that in the time-averaged picture, the optical forces from the pulsed laser are the same as those from a continuous laser when they have the same power density. The calculated force, together with the observed NP velocities and moving directions, leads to the confirmation of NP being moving in a nanobubble. In the manuscript, we first proposed this physical picture as a hypothesis and then used reasoning backed by EM calculations, Stokes' law and MD simulations to validate this hypothesis. Now with the additional study of the pulsed nature of the femtosecond laser in our simulations, we further confirmed the physical picture. We hope the revision is now more convincing.

Reviewer #2 (Remarks to the Author):

Optical manipulation of nanoparticles has been extensively investigated in recent years in many physics, such as molecular machinery, nanorobotics, and drug delivery. Especially, negative optical force is an interesting field in light-matter interaction. In this paper, the authors report optical pushing and pulling force of plasmonic nanoparticles with super-fast speeds. The numerical simulation and experimental results are structured well, which can fully support their claims. I think this article can be considered after addressing the following issues.

Comment 1. The authors give the conditions (geometrical windows) for obtaining negative optical force. And they explain it with negative light direction (kz). But what are the physical reasons for this counter-intuitive phenomenon? The authors could plot the energy flow around the nanoparticles. The singular points of Poynting vector (see Laser Photon. Rev. 9, 75(2015)) may play similar roles as the nano-bubble at the back side of the nanoparticles in your work.

Response: We thank the reviewer for this comment, which definitely helped us deepen the understanding of optical pulling and pushing forces on the nanoparticle in the nanobubble. We have carefully read the paper suggested by the reviewer and addressed this concern in the revised manuscript as the following:

From page 7, line 20 to page 8, line 17:

“The optical force on the NP in the nanobubble can be decomposed into two parts⁴⁰: one from the interaction between scattered fields themselves (F_z^{SS}), and the other from the interaction between the scattered field and the incident field (F_z^{Si}), and $F_z = F_z^{SS} + F_z^{Si}$. The sign and magnitude of each decomposed part can unveil which light field contributes dominantly to determining F_z . For a representative case of negative motion ($r_{nb}=130$ nm and $\theta=0^\circ$), it is found that F_z^{SS} is responsible for the optical pulling force, where F_z^{SS} is -1.254×10^{-12} N, but F_z^{Si} is only 4.99×10^{-13} N. Since F_z^{SS} is due to the net momentum the NP can gain by compensating the net scattered photon momentum leaving the NP, if the scattering is isotropic, F_z^{SS} should be zero. Thus, the negative F_z^{SS} implies that there is uneven radiative scattering from the NP. We have investigated the energy flow pattern of scattered photons with the Poynting vector and found that at the backside of the NP, there is a “saddle” singular point, which induces unusual energy flows around the NP (see the magenta triangle in **Fig. 2d**). The saddle point turns the energy flow scattered from the NP to the back side of the NP and leads the energy to flow around the “vortex” singular points located at the sides of the NP (see the cyan circles in **Fig. 2d**). It is this circulation of the scattered energy that allows the NP to possess negative momentum. Our result corresponds well to an analysis introduced in Ref.⁴¹ where a particle irradiated by a Bessel beam can receive an optical pulling force when the pairs of “saddle” and “vortex” singular points, which are located at the backside of the NP, redirect and focus the energy flow into the particle. Without a nanobubble, the saddle point

and the circulation of energy flow do not exist (see **Fig. 2e**), and the scattering is symmetric about the light polarization axis, leading to $F_z^{SS} = 0$. This indicates that the nanobubble is essential for forming the saddle singularity. We believe that the formation of a saddle point is due to the curvature of the nanobubble surface, as it works as an optical mirror to reflect the scattered light from the NP. We have also investigated the energy flow of scattered light for the representative case of positive motion ($r_{nb}=130$ nm and $\theta=180^\circ$). As the NP is now placed close to the right nanobubble surface, the saddle point is sandwiched between the NP and the nanobubble (see **Fig. 2f**). While there are still two vortex points which can circulate the energy flow to be redirected into the NP, the strength of circulation is much suppressed in comparison to the negative motion case. As a result, the magnitude of optical pulling force that the NP receives is much weaker than the pushing force, leading to a positive F_z^{SS} of 1.068×10^{-12} N.”

, in Reference:

“40. Salandrino, A., Fardad, S. & Christodoulides, D. N. Generalized Mie theory of optical forces. *J. Opt. Soc. Am. B* 29, 855-866 (2012).

41. Gao, D. et al. Unveiling the correlation between non-diffracting tractor beam and its singularity in Poynting vector. *Laser Photonics Rev.* 9, 75-82 (2015).”

, and in Fig. 2:

“

Figure 2| Optical force on an Au NP with a nano-bubble (a) Schematic of an Au NP with a nano-bubble. The Au NP (yellow sphere) consists of a 100 nm-diameter silica core and a 10 nm-thick Au shell. The nano-bubble with a radius of r_{nb} is attached to the surface (θ, φ) of the Au NP, where θ and φ are, respectively, the polar and the azimuthal angles in the polar coordinate with the origin (o) at the center of the Au NP. E_x , H_y , and k_z depict the electric field, the magnetic field, and the wavevector of the incident planewave, respectively. (b) The calculated optical force in the z-direction (F_z) as a function of r_{nb} . Here, $\theta = 0^\circ$, $\varphi = 0^\circ$ and the amplitude of E_x is $2.6 \times 10^6 \text{ V m}^{-1}$, corresponding to the laser in the experiment. The insets illustrate the schematic configurations of Au NP with a nano-bubble with different r_{nb} . (c) The calculated F_z as a function of θ and φ . Here, $r_{nb} = 130 \text{ nm}$ and the amplitude of E_x is $2.6 \times 10^6 \text{ V m}^{-1}$. On top of the contour, schematic configurations of Au NP with a nano-bubble as a function of θ are illustrated. (d-f) The directions and normalized magnitudes of Poynting vectors of scattered fields S_{sca} for the structures of (d) $r_{nb} = 130 \text{ nm}$ and $\theta = 0^\circ$, (e) without a nanobubble, and (f) $r_{nb} = 130 \text{ nm}$ and $\theta = 180^\circ$. Directions of F_z are also shown.”.

Comment 2. What is the force density distribution around the nanoparticles for the pulling and pushing case? And are there any significant difference of far-field scattering for these two cases? These may also help to understand the direction of the optical force.

In summary, I would recommend it for publication if my concerns are sufficiently considered.

Response: We thank the reviewer for this comment as well. We have investigated the optical force density and the far-field distribution of scattered fields for the cases of the positive and the negative motions and addressed them in the revised manuscript as the following:

On page 8, in lines 17 ~ 31:

“We have also investigated the optical stress tensor profile on the surface of the Au NP for the representative cases (see Supplementary Material Section SM6). Both cases show negative stress at the light-incoming side and positive stress at the rear side, and the sign of stress reverses as across the equator of the NP. For the negative motion case, it is observed that the negative stress around the side poles not only has a higher magnitude but also covers a larger area than those of the positive stress, thus leading to a net negative F_z^{SS} . We believe that the stronger negative stress is due to the strong energy circulation around the vortex singular point at the side poles of NP (see **Fig. 2d**). Meanwhile, for the positive motion case, the area and the magnitude of negative stress around the side poles are suppressed compared to that of the negative motion case, which yields a net positive F_z^{SS} . It is due to the weakened energy circulation as the saddle point is squeezed between the surfaces of the NP and the bubble (see **Fig. 2f**). We have also calculated the far-field scattering patterns (Supplementary Material Section SM7). It noted that the far-field scattering patterns do not intuitively correspond to the directions of optical forces on the NPs, which means that the optical force on the NP inside the nanobubble is the result of near-field scattering.”

, and in Supplementary Material Sections SM6 and SM7:

“Section SM6. Optical Stress Tensor Profile on the Surface of NP in Nanobubble

Figure S7. The normalized stress tensor profile from the scattered fields of cases with (a) $r_{nb}=130$ nm and $\theta=0^\circ$ and (b) $r_{nb}=130$ nm and $\theta=180^\circ$. Here, \mathbf{n} depicts the normal vector at the surface of Au NP.

Section SM7. Far-Field Scattering Pattern of NP in Nanobubble

Figure S8. The far-field scattering patterns of the cases with (a) $r_{nb}=130$ nm and $\theta=0^\circ$ and (b) $r_{nb}=130$ nm and $\theta=180^\circ$. Here, the normalized scattering electric field amplitude ($|E_{ff}|/|E_{ff}|_{max}$) in the far-field domain is plotted as a function of the polar angle in the x-z plane at $y=0$. $|E_{ff}|_{max}$ is the maximum of $|E_{ff}|$ at the polar angle of 180° . The location of NP being encapsulated by the nanobubble is at the center of the polar coordinate. Each circle in the polar coordinate corresponds to an isovalue of $|E_{ff}|/|E_{ff}|_{max}$.

Reviewer #3 (Remarks to the Author):

The authors reported the design and optically-driven motions of ultrafast nanoswimmers via a new mechanism of bubble wrapping. There are certainly innovative and exciting aspects of the work. However, the paper needs to be significantly revised to make sure some of their conclusions and terminology are correctly described, especially the key mechanism of optical supercavitation that the authors argued. Detailed suggestions are as below:

Comment 1. For their discussions on Figure 1b and 1c, the ultrafast motions of the nanoswimmers: their measurement of the speed is based on movies like Supplementary Video 1. However, Supplementary Video 1 as it is now has both particles that are moving superfast and those are only moving locally. Can the authors clearly single out/label the particles based on which they argue about the superfast motions, and detail how the average speed was calculated (instantaneous or averaged speed?). Also explain why there are heterogeneity in the particle motions – is that due to synthetic heterogeneity of the particles themselves or just because the light intensity is uneven? There are some discussions about this on Page 8, which don't alleviate the concern that the light pulse is not a truly robust and reproducible method to drive the particles. Moreover, the particle concentration in Supplementary Video 1 is very high, which can make it difficult to track single particle motions (i.e., how to link the particles across frames as the same particle). Do the authors have low particle concentration movies, without ambiguities on tracking?

Response: As there are a number of comments in **Comment 1**, we addressed them one-by-one as the following.

Comment 1.1. For their discussions on Figure 1b and 1c, the ultrafast motions of the nanoswimmers: their measurement of the speed is based on movies like Supplementary Video 1. However, Supplementary Video 1 as it is now has both particles that are moving superfast and those are only moving locally. Can the authors clearly single out/label the particles based on which they argue about the superfast motions, and detail how the average speed was calculated (instantaneous or averaged speed?).

Response: We thank the reviewer for this comment. Yes, we can label the ballistic NPs. To address this concern, two new video which labels/highlights the ballistic NPs are provided as **Supplementary Video 2 (for the positive motion) and Supplementary Video 3 (for the negative motion)** with the revised manuscript. Actually, we also plotted their trajectories in **Figs. 1d and 1e** in the original manuscript.

Since we can track the position of the ballistic NPs as a function of time, we can calculate their instantaneous speeds as a function of time. As can be seen from the highlighted videos and the trajectories in Figs. 1d and 1e, we can see that the NPs can travel for tens or even over one hundred

micrometers with their speeds. For a set of the observed representative negative motions (or positive motions), we also averaged their speeds to report the averaged speed for each direction.

We have clarified the above points in the revised manuscript as the following:

On page 4, in lines 23 ~ 29:

“We single out several representative cases of ballistic NPs moving in either direction (see examples in Supplementary Video 2 for the positive motion and Supplementary Video 3 for the negative motion), track their positions as a function of time, and calculate their instantaneous speeds. We find that the maximum speed for the positive motion is $336,000 \mu\text{m s}^{-1}$ and that for the negative motion is $245,000 \mu\text{m s}^{-1}$ (see Figs. 1b and 1c). We also find that the average speed of the observed representative positive motions ($204,000 \mu\text{m s}^{-1}$) is higher than that of the negative motion ($109,000 \mu\text{m s}^{-1}$).”

Comment 1.2. Also explain why there are heterogeneity in the particle motions – is that due to synthetic heterogeneity of the particles themselves or just because the light intensity is uneven? There are some discussions about this on Page 8, which don't alleviate the concern that the light pulse is not a truly robust and reproducible method to drive the particles.

Response: We thank the reviewer for this comment. First, we are sure that our pulsed laser is strictly mode-locked and produce uniform pulses as calibrated by the manufacturer. The heterogeneity in the particle motions (i.e., some move in positive direction, some in negative direction and some drift slowly) can actually be from many different factors, including the heterogeneity of the particles, location of the particles, the inherent stochastic nature of bubble dynamics (nucleation and growth), and the relative position of NP inside the bubble. As discussed in the manuscript, the ballistic motion really requires the NPs to be encapsulated by a nanobubble generated from photo-thermal heating. In experiments, ballistic NPs are mostly observed within $\sim 300 \mu\text{m}$ at either side of the focal plane since the laser intensity is sufficiently high in this region (note we have a weakly focused Gaussian beam as mentioned in the manuscript). NPs outside of this region are mainly seen drifting due to thermal convection perpendicular to the laser beam direction, which can be seen from the videos. However, even within this region, not every NP is guaranteed to move ballistically since bubble generation is inherently statistical and the probability may depend on factors like defects on the NP surface and geometry of the NP (defects and sharp corners may change the nucleation threshold like in classical pool boiling). If a nanobubble is generated and the NP is encapsulated, moving forward or backward still depends on the relative position of the NP inside the bubble (as discussed in Fig. 2).

We have clarified the above points in the revised manuscript as the following:

On page 10, in lines 20 ~ 27:

“This is because the laser intensity is sufficiently high in this region. NPs outside of this region are mainly seen drifting due to thermal convection perpendicular to the laser beam direction. However, even within this region, not every NP is guaranteed to move ballistically since bubble generation is inherently statistical and the probability may depend on factors like defects on the NP surface and geometry of the NP (e.g., defects and sharp corners are known to change bubble nucleation threshold⁴⁵). If a nanobubble is generated and the NP is encapsulated, moving forward or backward still would depend on the relative position of the NP inside the bubble (as discussed in Fig. 2).”.

, and in Reference:

“45. Li, C. et al. Nanostructured copper interfaces for enhanced boiling. *Small* **4**, 1084-1088 (2008).”.

Comment 1.3. Moreover, the particle concentration in Supplementary Video 1 is very high, which can make it difficult to track single particle motions (i.e., how to link the particles across frames as the same particle). Do the authors have low particle concentration movies, without ambiguities on tracking?

Response: We thank the reviewer for this comment. Yes, we do have videos from lower NP concentration suspensions. Supplementary Video 4 and Video 5 with the revised manuscript, which show the same ballistic NPs in low particle concentrations. We can see that the negative motion (in Supplementary Video 4) and the positive motion (in Supplementary Video 5) are clearly shown without ambiguities on tracking their locations as a function of time.

We have addressed this concern in the revised manuscript as the following:

On page 5, at line 15 ~ 19:

“At lower NP concentrations, we also find ballistic NPs with both negative (Supplementary Video 4) and positive motions (Supplementary Video 5), suggesting that the ballistic movements should not be the result of the environment (e.g., scattered light from surrounding NPs or thermalization of water due to NP heating).”.

Comment 2. The discussions on the three possible optical forces on page 5 in the main text: can the authors make them more quantitative? How small is “small”? Currently the discussions are very qualitative. One can only rule out these effects after quantitatively comparing them with the supercavitation effects.

Response: We thank the reviewer for this comment. We mentioned three possible forces, including optical pressure force, optical gradient force and photo-thermal gradient force. However, the observed ballistic motions of the NPs immediately eliminated the possibility of gradient forces (including optical and thermal gradient forces), since the NPs can cross the focal plane (black arrow in Figs. 1d and 1e) regardless of their moving directions. In addition, we can see that to the left (or right) side of the focal plane, ballistic NPs can move in both directions. These indicate that the ballistic NPs are not driven by any gradient (optical or photothermal) forces, since such forces are symmetric about the focal plane, which should converge the NPs from both sides of the focal plane towards it. To address this issue in a more quantitative manner, we performed a finite element analysis and have addressed this concern in the revised manuscript as the following:

On page 5, in lines 10 ~ 15:

“We have also calculated the optical force field on the NP under the Gaussian beam used in the experiment (see Supplementary Materials Section SM1). We find that there is only positive force on the NP and the amplitude is almost symmetric to the focal plane. If gradient optical force is significant, it should have broken such a symmetry since it changes sign across the focal plane. We can thus infer that the gradient optical force is negligible compared to the dissipative optical forces.”

, and in Supplementary Material Section SM1:

“Section SM1. Optical Force Field on a SiO₂-Au Core-Shell NP under the Gaussian Beam

Figure S1| The calculated F_z on a SiO₂-Au core-shell NP along the central axis (z-direction) of the Gaussian beam with the beam waist of 6 μm and the intensity of 12 mW μm⁻² at the focal plane (z = 0).”

Comment 3. On the optical force calculation presented on page 6: It is not clear why a nanobubble of a size larger than the NP suggests complete enveloping; the nanoparticles can have a big

nanobubble attached only partially on the NP surface, which is the mechanism for numerous literatures on self-propelled micron-sized particles. It is not clear why the bubble would want to completely wet the particle surface and create this frictionless environment.

Response: We thank the reviewer for this comment. As far as we know, for the self-propelled micro/nano particles, the particles are shaped to be asymmetric (e.g., Janus sphere or conical tube) and designed to repel a chemically formed bubble so that they can receive a recoil force when a bubble detaches from the particle. For this study, upon the laser excitation, a nanobubble can be nucleated from a certain location on the surface of an Au NP⁴. However, after the nucleation, the nanobubble can fully encapsulate the NP if the fluence of the femtosecond pulse is higher than a threshold value¹². For the NP with the SPR peak at 800 nm, it has been reported that the threshold is around $\sim 7 \text{ mJ cm}^{-2}$ (Ref.¹²), and we have used a laser with fluences of $9 \sim 15 \text{ mJ cm}^{-2}$. We could also estimate that the temperature of Au NP under the pulsed light can be up to 850 K, which is much higher than the critical temperature of water (please see the response to **Comment 1** of Reviewer 1). There have also been reports showing that the temperature of the NP subject to the SPR laser heating reaching $> 1000 \text{ K}$, which is not possible without a nanobubble fully encapsulating the NP³. Thus, it should be very difficult to achieve partial de-wetting of the NP surface with a nanobubble, especially considering that the temperature of the NP is so high and keeps receiving laser energy.

To clarify this concern, we have added the following sentences in the revised manuscript:

On page 6, at line 29 to page 7, at line 6:

“Our assumption of the nanobubble nucleation and the encapsulation of NP is based on experimental studies, where Fu et al.³⁰ showed that upon laser excitation, a nanobubble can nucleate at a certain location on the Au NP surface and Lachaine et al.³² showed that the nanobubble can fully encapsulate the NP if the fluence of the femtosecond laser pulse is higher than a threshold. For the NP used in this study, the threshold is around $\sim 7 \text{ mJ cm}^{-2}$ (Ref.³²), and our laser fluences are of $9 \sim 15 \text{ mJ cm}^{-2}$. The encapsulation of NP by the nanobubble is also intuitive given the very high temperature of the NP ($> 850 \text{ K}$), which is above the water critical temperature and can instantaneously evaporate liquid in contact with the hot NP surface.”

, and in Reference:

“30. Fu, X., Chen, B., Tang, J. & Zewail, A. H. Photoinduced nanobubble-driven superfast diffusion of nanoparticles imaged by 4D electron microscopy. *Sci. Adv.* **3**, e1701160 (2017).

32. Lachaine, R. et al. Rational design of plasmonic nanoparticles for enhanced cavitation and cell perforation. *Nano Lett.* **16**, 3187-3194 (2016).”

Comment 4. How does photothermal effect play a role now that the light is at the plasmonic resonance wavelength of the gold NP?

Response: We thank the reviewer for this question. The photothermal effect when the light is at the SPR peak of the NP is the key to the superfast motion of the NP. The photothermal conversion at the NP SPR can increase the temperature of the NP to values much higher than the critical temperature (647 K) of water (Please see the response to **Comment 1** of Reviewer 1). The hot NP thus can instantly evaporate water molecules and can be fully encapsulated by vapor (see response to the last comment). In addition, the NP is continuously excited by the laser, and thus can keep the high temperature.

To clarify this concern, we have added the following sentences in the revised manuscript:

On page 6, at line 5 ~ 11:

“Our heat transfer analysis also estimates that the plasmonic heating can rise the temperature of an NP inside a nanobubble to ~850 K (see Supplementary Material Section SM3), which is much higher than the critical temperature of water (647 K). This calculated temperature from our simple model is also in reasonable agreement with that inferred from experiments (~1000 K)³⁴. The hot NP thus can instantly evaporate water molecules, like the Leidenfrost effect, and can be fully encapsulated by vapor³⁰⁻³³. As long as the NP is illuminated by the laser, the NP can keep the high temperature.”

, and in Reference:

“30. Fu, X., Chen, B., Tang, J. & Zewail, A. H. Photoinduced nanobubble-driven superfast diffusion of nanoparticles imaged by 4D electron microscopy. *Sci. Adv.* **3**, e1701160 (2017).

31. Lukianova-Hleb, E., Volkov, A. N. & Lapotko, D. O. Laser pulse duration is critical for generation of plasmonic nanobubbles. *Langmuir* **30**, 7425-7434 (2014).

32. Lachaine, R. et al. Rational design of plasmonic nanoparticles for enhanced cavitation and cell perforation. *Nano Lett.* **16**, 3187-3194 (2016).

33. Metwally, K., Mensah, S. & Baffou, G. Fluence threshold for photothermal bubble generation using plasmonic nanoparticles. *J. Phys. Chem. C* **119**, 28586-28596 (2015).

34. Hodak, J. H., Henglein, A., Giersig, M. & Hartland, G. V. Laser-induced inter-diffusion in AuAg Core-Shell Nanoparticles. *J. Phys. Chem. B* **104**, 11708-11718 (2000).”

Comment 5. In their last discussion on gold nanorods, again the explanation of the direction dependence and the lack of negative motions is very hand-wavy. The authors need to articulate in a logically way, with supplementary notes as appropriate, using experimental data to verify their discussions, instead of just making unverified statements. For example, for the alignment argument: can the authors show what is the orientation of gold nanorods as they exhibit positive motions? Parallel or perpendicular to the motion direction?

Response: We thank the reviewer for this comment. Due to the anisotropy of the Au nanorods (NRs), the SPR depends on their orientation with respect to the light polarization. This makes excitation at SPR to have a lower probability than isotropic Au NPs. As discussed in the response to the last comment, SPR excitation is essential to supercavitation due to the significant heat needed. Further considering the heterogeneity issues discussed in our response to **Comment 1.2**, the probability to observe ballistic motion is even lower. That is why we could hardly observe any ballistic motion except one case. To address the reviewer’s concern, we have calculated the optical properties of Au NR in the nanobubble and addressed them in the revised manuscript as the following:

In Supplementary Material Section SM10:

“**Section SM10. Super-fast Ballistic Movement of Au Nanorods**”

Figure S10 | (a) The calculated scattering cross-section of an Au nanorod (NR) when it is aligned with the x, y, or z-axis. Only when the Au NR is aligned with the x-axis the SPR is possible. (b) Dark-field optical images of a ballistic Au NR with a positive motion. (c) The calculated optical force in the z-direction (F_z) on the x-oriented Au NR as a function of the radius of nano-bubble (R_{nb}). Here, the amplitude of E_x is $2.6 \times 10^6 \text{ V m}^{-1}$. The insets illustrate the schematic configurations of Au NR with a nano-bubble with different R_{nb} .

We perform an experiment with Au nanorod (NR) to check whether the observed ballistic movement is potentially generalizable. The Au NR has a length of 48 nm and a width of 12 nm. The Au NR in water can have an SPR peak at the wavelength (λ) of 800 nm when the axis of NR is parallel to the polarization direction of the incident light, which is clearly shown in **Fig. S10a**. It indicates that the Au NR can be intensely excited only when aligning with the field direction of a 800 nm laser. The Au NRs are dispersed in water with the number density of $1.3 \times 10^{17} \text{ \#/m}^3$. We use the same experimental setup described in **Fig. 1a** in the main text to observe NR motion. A femtosecond pulsed laser with a power of $\sim 1 \text{ W}$ passes through a $20\times$ objective lens, introducing a Gaussian beam in the Au NR-water suspension. Since the SPR characteristic of Au NR is anisotropic due to the shape of the rod, only the Au NR whose longitudinal axis is well aligned with the direction of incident electric field can have the SPR and be heated to create supercavitation. This implies that there is a low probability to observe ballistic Au NRs, because slight Au NR misalignment with the electric fields may interrupt the heating and thus supercavitation. Although rare, in **Fig. S10b**, we could still observe a glowing dot moving over a distance of $\sim 13 \text{ \mu m}$ within 1 ms almost perfectly along the beam propagating direction (see also Supplementary Video 6). This ballistic Au NR shows the normalized speed of 260,000 body-length s^{-1} , which is $\sim 10^2$ - 10^5 times faster than the reported nano/micro swimmers driven by optical forces^{13,14}. The magnitude of the optical force has the range of $1 \times 10^{-13} \text{ N} \sim 7 \times 10^{-13} \text{ N}$ and monotonically decreases as the size of nanobubble increases, as calculated using EM wave simulation (**Fig. S10c**). Balancing the averaged magnitude of optical forces with the Stokes' drag force can yield a dynamic viscosity of $4.4 \times 10^{-5} \text{ kg m}^{-1}\text{s}^{-1}$, which is much lower than that of liquid water and on the same order of magnitude as that of vapor. This result demonstrates that the observed ballistic NP movement can be potentially generalized to other kinds of plasmonic NPs with different geometry, composition and dimensions as long as the SPR heating can be excited.”

, and in the main text on page 11 in line 20 ~ 23:

“With the calculated optical force ($1 \times 10^{-13} \text{ N} \sim 7 \times 10^{-13} \text{ N}$) on the Au NR (see Supplementary Materials Section SM10), Stokes' law implies a dynamic viscosity of $4.4 \times 10^{-5} \text{ kg m}^{-1}\text{s}^{-1}$, which is much lower than that of liquid water and on the same order of magnitude as that of vapor.”.

Reference

1. Gao, W., Sattayasamitsathit, S. & Wang, J. Catalytically propelled micro-/nanomotors: How fast can they move? *Chemical Record* **12**, 224–231 (2012).
2. Xuan, M. *et al.* Near Infrared Light-Powered Janus Mesoporous Silica Nanoparticle Motors. *J. Am. Chem. Soc.* **138**, 6492–6497 (2016).
3. Hodak, J. H., Henglein, A., Giersig, M. & Hartland, G. V. Laser-Induced Inter-Diffusion in AuAg Core-Shell Nanoparticles. *J. Phys. Chem. B* **104**, 11708–11718 (2000).
4. Fu, X., Chen, B., Tang, J. & Zewail, A. H. Photoinduced nanobubble-driven superfast diffusion of nanoparticles imaged by 4D electron microscopy. *Sci. Adv.* **3**, e1701160 (2017).
5. Boulais, É., Lachaine, R. & Meunier, M. Plasma-Mediated Nanocavitation and Photothermal Effects in Ultrafast Laser Irradiation of Gold Nanorods in Water. *J. Phys. Chem. C* **117**, 9386–9396 (2013).
6. Hinds, E. A. & Barnett, S. M. Momentum exchange between light and a single atom: Abraham or Minkowski? *Phys. Rev. Lett.* **102**, (2009).
7. Scalora, M. *et al.* Radiation pressure of light pulses and conservation of linear momentum in dispersive media. *Phys. Rev. E - Stat. Nonlinear, Soft Matter Phys.* **73**, (2006).
8. Mansuripur, M. & Zakharian, A. R. Maxwell's macroscopic equations, the energy-momentum postulates, and the Lorentz law of force. *Phys. Rev. E - Stat. Nonlinear, Soft Matter Phys.* **79**, (2009).
9. Ellingsen, S. Å. Theory of microdroplet and microbubble deformation by Gaussian laser beam. *J. Opt. Soc. Am. B* **30**, 1694 (2013).
10. Nieminen, T. A. *et al.* Optical tweezers: Theory and modelling. *J. Quant. Spectrosc. Radiat. Transf.* **146**, 59–80 (2014).
11. Preez-Wilkinson, N. du, Stilgoe, A. B., Alzaidi, T., Rubinsztein-Dunlop, H. & Nieminen, T. A. Forces due to pulsed beams in optical tweezers: linear effects. *Opt. Express* **23**, 7190 (2015).
12. Lachaine, R., Boutopoulos, C., Lajoie, P.-Y., Boulais, É. & Meunier, M. Rational Design of Plasmonic Nanoparticles for Enhanced Cavitation and Cell Perforation. *Nano Lett.* **16**, 3187–3194 (2016).
13. Königer, A. & Köhler, W. Optical Funneling and Trapping of Gold Colloids in Convergent Laser Beams. *ACS Nano* **6**, 4400–4409 (2012).
14. Kajorndejnukul, V., Ding, W., Sukhov, S., Qiu, C.-W. & Dogariu, A. Linear momentum increase and negative optical forces at dielectric interface. *Nat. Photonics* **7**, 787–790 (2013).

Reviewers' comments:

Reviewer #1 (Remarks to the Author):

The authors have amended the manuscript and, to certain degree, have answered some of the previous remarks.

The authors decided to drop the word swimmers from their title. However, the argument about the size of those particles that are shown in figure S2 is unconvincing. It is hard to conclude that "...their sizes are well-defined (diameter of 120 nm)" given that the scale bar is 10 μ m (!) in that figure.

The justification for the use of a continuous wave approximation is appropriate.

In response to my comment, the authors introduce additional information such as calculation of the temperature distribution in and around the nanoparticle. Based on this, they argue that the mechanism of bubble formation is not important for their observations because "the nanobubble can only move with a speed more than two orders of magnitude slower...". But the point was not that the NP and the bubble moves independently! Together with the nanoparticle, they form a complex dielectric structure and the electromagnetic field interacts with this entire structure. The electromagnetic force should be evaluated for the entire structure. Moreover, a compelling argument about what they call supercavitation should involve the time scales for heating, evaporation, electrostriction, etc.

The current treatment is still not satisfactory for a complex situation as also proven by the ad-hoc situation depicted in Fig 2. So far, this picture of a small particle encapsulated in a nanobubble moving forward or backward is not supported by the electromagnetic arguments presented in this manuscript.

Indeed, the authors have improved the revised manuscript but the intriguing consequences of focusing a femtosecond laser into a colloidal suspension of metallic nanoparticles are not yet satisfactorily explained.

Reviewer #2 (Remarks to the Author):

The authors have properly answered all of my concerns. I totally recommend publication of the paper.

Reviewer #3 (Remarks to the Author):

The authors have satisfyingly addressed all of my comments. However, their response to comment 1.2 basically confirms my concern that the heterogeneity of the ballistic motions (some particles move, some don't, some in opposite directions) is indeed unavoidable after all. This fact put the robustness of the work in question.

Reviewer 1:

The authors have amended the manuscript and, to certain degree, have answered some of the previous remarks.

Comment 1: The authors decided to drop the word swimmers from their title. However, the argument about the size of those particles that are shown in figure S2 is unconvincing. It is hard to conclude that "...their sizes are well-defined (diameter of 120 nm)" given that the scale bar is 10um (!) in that figure.

Response: The particles have well-defined sizes with a diameter (c.a. 120 nm), and the product was from a prominent vendor which has historically provided consistent quality. We provide the inset in Fig. S2b to show an SEM to verify the size. What we intended to show in Fig. S2 in our last response was that the particles did not cluster, which would thus address the original question from Reviewer 1 that we could not know the particle size from a CCD camera. We have now added an additional SEM image to Fig. S2 to unambiguously show the size.

Figure S2| (a) An experimental setup to deposit the ballistic NPs with negative motion on a quartz substrate. An objective lens (10×, beam waist ~20 μm) was used to focus the femtosecond laser with a power of 690 mW on the substrate surface. (b) The collected ballistic NPs on the quartz substrate visualized using SEM. The inset is the magnified SEM image of an Au NP on the quartz.

We have further clarified what we would like to show in Fig. S2 by modifying the relevant sentence in the manuscript:

“Additionally, when we deposited the ballistic NPs on a quartz substrate and then characterized their geometrical configurations using a scanning electron microscope, it is found that the deposited NPs are isolated single NPs **without clustering** (see Supplementary Materials Section SM2).”

Comment 2: The justification for the use of a continuous wave approximation is appropriate.

Response: We thank the reviewer for this comment.

Comment 3: In response to my comment, the authors introduce additional information such as calculation of the temperature distribution in and around the nanoparticle. Based on this, they argue that the mechanism of bubble formation is not important for their observations because “the nanobubble can only move with a

speed more than two orders of magnitude slower...”. But the point was not that the NP and the bubble moves independently! Together with the nanoparticle, they form a complex dielectric structure and the electromagnetic field interacts with this entire structure. The electromagnetic force should be evaluated for the entire structure. Moreover, a compelling argument about what they call supercavitation should involve the time scales for heating, evaporation, electrostriction, etc.

Response: The critic is that we should have used the electromagnetic (EM) force (i.e., optical force) on the bubble/NP structure as a whole, instead of only the force on NP, to explain the ballistic motion of the NP. However, the reviewer is not correct! We have discussed in the original manuscript, and re-emphasized in the last response letter, that the bubble boundary is extended *in-situ* during the NP movement since the laser excited NP has a very high temperature (> 800 K) and can instantaneously evaporate water. It is like the Laidenfrost effect, where a hot object instantaneously evaporates the close-by water to provide a vapor cushion between the hot body and the surrounding liquid. Simply put, the NP is moving in a gaseous environment (low-friction) created instantaneously by its own heat, not that the bubble was created first and then the same bubble move together with the NP. This was illustrated and verified by our MD simulations in the paper (see Fig. 3c in manuscript and the movie of our MD result: <https://www.youtube.com/watch?v=vaCRx9C3ox8>). To make this phenomena easy to understand, we have also made the analogy of our observation to the supercavitating torpedo (<https://www.militaryaerospace.com/power/article/16726685/is-world-ready-for-an-undersea-missile-supercavitating-torpedo-offers-speed-of-230-miles-per-hour>), where an on-board gas generator on the nose of torpedo creates a bubble to wrap around the torpedo so it moves in a low-friction gaseous environment instead of viscous liquid water (see the left picture below). There, the bubble boundary was extended *in-situ* with the continuous gas release from the gas generator as the torpedo moves. The recently reported Cav-x also share the same mechanism (<https://newatlas.com/military/dsg-cavx-supercavitating-underwater-bullets/>). The difference in our case is that we used the laser heating of the plasmonic NP to extend the bubble boundary via instantaneous evaporation instead of gas generation (see the right schematic below). We are not trying to use these cartoon pictures to explain the physics, but just want to make our point crystal clear to avoid any misunderstanding. Our manuscript has backed our argument with calculations and simulations. We think the phenomena is fascinating, and our EM modeling, calculated optical force, observed NP speed, and MD simulations piece together perfectly to explain the physics (including the fast speed and optical pulling phenomena).

[redacted]

Given the above explanation, we hope the reviewer can agree that the force on the bubble is not relevant because the hot NP can always instantaneously extend the boundary of the bubble wherever it moves to ensure it is encapsulated in a low-friction gaseous environment. In this case, only the EM force on the NP matters. We emphasize again: it is not the same original bubble that moves with the NP. Think about this scenario: if in our observation it is the original bubble that moves together with the NP, the optical force on the bubble (6.0×10^{-13} N from EM calculation) would have to drive the observed movement (see our manuscript, which excluded all other forces as possible drivers). The speed, according to Stokes law, would be two orders of magnitude smaller than our experimental observation since the bubble is immersed in liquid water, which is two orders of magnitude more viscous than vapor phase. Also, if they move together, the observed optical pulling could never have happened – there is no mechanism to enable the negative force on the bubble.

For convenience, we have appended our response to the reviewer's previous comment 2 at the end, where we have highlighted the text we clarified the above points.

In terms of timescales, the instantaneous evaporation happens in sub-nanoseconds as shown in the MD simulation. Plasmonic heating should be at the time scale of picosecond or sub-picosecond, which is a charge-phonon coupling process. These timescales are much shorter than that of the NP movement, and thus can be regarded as instantaneous. And as we discussed in our response, electrostriction should not be important since the bubble boundary is extended by evaporation.

Comment 4: The current treatment is still not satisfactory for a complex situation as also proven by the ad-hoc situation depicted in Fig 2. So far, this picture of a small particle encapsulated in a nanobubble moving forward or backward is not supported by the electromagnetic arguments presented in this manuscript.

Response: The configuration space (relative position of NP inside bubble) is infinite and one can only analyze representative cases to verify the hypothesized mechanism of positive and negative optical forces. However, the representative cases we studied explained our observation (positive and negative motions) very well if one considers the fact that the hot NP can instantaneously extend the bubble boundary.

Reviewer 3:

Comment: The authors have satisfyingly addressed all of my comments. However, their response to comment 1.2 basically confirms my concern that the heterogeneity of the ballistic motions (some particles move, some don't, some in opposite directions) is indeed unavoidable after all. This fact put the robustness of the work in question.

Response: We do not believe this is a robustness issue, but rather the reflection of the stochastic nature of the phenomena. We emphasize that we can reproduce the observation every time we do the experiment, and we have presented a few movies with the manuscript which all show ballistic motion in both positive and negative directions. First, the bubble formation is stochastic in nature. The relative position of NP inside the bubble is also stochastic. Since the ballistic motion depends on the generation of the bubble, and the NP motion direction depends on the relative position of NP inside the bubble, the observation is supposed to be stochastic in nature, *i.e.*, some moving in the positive direction, some in the negative direction, and some don't move ballistically when no bubble is formed. The stochastic feature is intrinsic, and it is indeed unavoidable, as it is part of the physics. We have modified the language on page 10 to make this point clearer:

“... However, even within this region, not every NP is guaranteed to move ballistically since bubble generation is **stochastic in nature** and the probability may depend on factors like defects on the NP surface and geometry of the NP (*e.g.*, defects and sharp corners are known to change bubble nucleation threshold⁴⁵). If a nanobubble is generated and the NP is encapsulated, moving forward or backward still would depend on the relative position of the NP inside the bubble (as discussed in Fig. 2). ...”

Appendix: Response to Reviewer 1's previous comment 2

Comment 2. Nevertheless, the reported experiments were carefully conducted. An independent pump-probe experiment suggests that gas bubbles are sometimes formed around the metallic particles and it is argued that this is the reason for the high velocities recorded. This is an intriguing hypothesis and, to support it and to understand some of its implications, the authors perform calculations of the optical force. Unfortunately, this part is rather sketchy.

First of all, it is not clear how the Maxwell stress tensor is calculated. There two types of interfaces here: AU-gas and gas-liquid. Which ones are included? I understand that the results presented refer to the force acting on NP but how about the action on the bubble itself? In this respect, the authors can consult, for instance, J. Opt. Soc. Am. B 30(6), 1694 (2013).

Response: We thank the reviewer for these questions. Yes, the optical force was calculated by integrating the Maxwell stress tensor over the Au NP/vapor interfaces. So, it was the force acting on the Au NP. Before discussing the optical force on the nanobubble, we would like to first show the rise of temperature of the NP due to the plasmonic heating from the laser. We have calculated the time-averaged optical absorptions and estimated how the absorbed energy can heat up the NP using the finite element method to solve the continuity, momentum and heat transfer equations of the NP-in-nanobubble system in water. The temperature profile of the system is shown in **Fig. S3 in the revised Supplementary Materials**, where the NP temperature can reach ~850 K, much higher than the critical temperature of water (647 K). This calculated temperature from our simple heat transfer model is also not far from that inferred from experiments (~1000 K)³. **At such a high temperature, water molecules come in contact with the hot NP can be instantly vaporized into the gaseous phase – like the Leidenfrost effect. While the focus of this study is not on the mechanism of nanobubble formation, there have been a number of reports in the literature showing that nanobubble can be generated instantly by laser irradiation at the NP SPR^{4,5} as we have cited in the manuscript. In addition, we have ultra-fast pulsed laser with a repetition rate of 80.7 MHz. Our calculation has shown that the temperature would only decrease by a few degrees before it is heated by another pulse (see calculation in the original Supplementary Materials). It is thus reasonable to say that wherever the hot NP is moved to by the optical force, it can instantly evaporate water molecules. Thus, a new interface of nanobubble at the front of the NP can be developed wherever the hot NP moves to, i.e., it is not the same bubble moving together with the NP, but the boundary of the bubble is extended by instantaneous evaporation. This was also confirmed by our MD simulations in the paper.**

We have addressed this concern in the revised manuscript as the following:

On page 6, in lines 5 ~ 11:

“Our heat transfer analysis also estimates that the plasmonic heating can rise the temperature of an NP inside a nanobubble to ~850 K (see Supplementary Material Section SM3), which is much higher than the critical temperature of water (647 K). This calculated temperature from our simple model is also in reasonable agreement with that inferred from experiments (~1000 K)³⁴. The hot NP thus can instantly evaporate water molecules, like the Leidenfrost effect, and can be fully encapsulated by vapor³⁰⁻³³. As long as the NP is illuminated by the laser, the NP can keep the high temperature.”

, in Reference:

“30. Fu, X., Chen, B., Tang, J. & Zewail, A. H. Photoinduced nanobubble-driven superfast diffusion of nanoparticles imaged by 4D electron microscopy. *Sci. Adv.* **3**, e1701160 (2017).

31. Lukianova-Hleb, E., Volkov, A. N. & Lapotko, D. O. Laser pulse duration is critical for generation of plasmonic nanobubbles. *Langmuir* **30**, 7425-7434 (2014).

32. Lachaine, R. et al. Rational design of plasmonic nanoparticles for enhanced cavitation and cell perforation. *Nano Lett.* **16**, 3187-3194 (2016).

33. Metwally, K., Mensah, S. & Baffou, G. Fluence threshold for photothermal bubble generation using plasmonic nanoparticles. *J. Phys. Chem. C* **119**, 28586-28596 (2015).

34. Hodak, J. H., Henglein, A., Giersig, M. & Hartland, G. V. Laser-induced inter-diffusion in AuAg Core-Shell Nanoparticles. *J. Phys. Chem. B* **104**, 11708-11718 (2000).”.

and in Supplementary Material Section SM3:

“Section SM3. Temperature of Plasmonic Au NP in a Nanobubble

Figure S3 | The calculated temperature of the NP-in-nanobubble system when the amplitude of the electric field of an incident planewave is 2.6×10^6 V/m, corresponding to the experimental laser condition.”.

In the meantime, the optical forces on the nanobubble, given by the time-averaged Maxwell stress tensor, are usually positive with magnitudes comparable to those on the NPs. The calculated values are around 6.0×10^{-13} N with slight variations depending on where the NP is inside the bubble. With this magnitude of force, a nanobubble itself can only move with a speed more than two orders of magnitude slower (Stokes’s law yields a speed of $\sim 410 \mu\text{m s}^{-1}$) than the ballistic NPs, as the nanobubble is dragged by the viscosity of

liquid water. However, as discussed above, the movement of the bubble is not very important as it is the instantaneous evaporation that keeps the NP inside a bubble, instead of the synchronization between the dynamical motions of the NP and the bubble. The paper referenced by the reviewer also mentioned electrostriction forces. It is noted that the deformation of nanobubble due to the electrostriction force should not be important as a new interface of nanobubble is formed by the hot moving NP.

We have addressed this concern in the revised manuscript as the following:

From page 10, line 7 ~ 15:

“We also note that the bubble can experience optical forces as well, but they are usually positive with amplitude calculated to be around 6.0×10^{-13} N. However, with such forces, the nanobubble can only move with a speed more than two orders of magnitude slower (e.g., Stokes’s law yields a speed of $\sim 410 \mu\text{m s}^{-1}$) than the NPs. Therefore, it should be the instantaneous evaporation that keeps the NP inside a gaseous environment, instead of the synchronization between the dynamical motions of the NP and the bubble. We mention that there is also electrostriction⁴⁴ forces that can deform the nanobubble, but in our scenario it should not be important as new interfaces of nanobubble is formed by the hot moving NP.”

, and in Reference:

“44. Ellingsen, S. A. Theory of microdroplet and microbubble deformation by Gaussian laser beam. *J. Opt. Soc. Am. B.* **30**, 1694-1710 (2013).”.

REVIEWERS' COMMENTS:

Reviewer #2 (Remarks to the Author):

The authors have adequately responded to my comments and the manuscript can be accepted for publication.

Reviewer #3 (Remarks to the Author):

I have reviewed the authors' responses to both reviewers and I am convinced that the paper has now passed the bar for publication in Nature Communications.

Dear Editor & Reviewers,

We appreciate the time and effort you put to the editorial and reviewing processes regarding our manuscript (NCOMMS-19-16020B-Z), “Ballistic Supercavitating Nanoparticles Driven by Single Gaussian Beam Optical Pushing and Pulling Forces”, by Eungkyu Lee, Dezhao Huang, and Tengfei Luo. We have included our point-by-point responses below.

Reviewer #2 (Remarks to the Author):

Comment 1. The authors have adequately responded to my comments and the manuscript can be accepted for publication.

Response: Thank you very much for the review and supporting our manuscript for publication!

Reviewer #3 (Remarks to the Author):

Comment 1. I have reviewed the authors' responses to both reviewers and I am convinced that the paper has now passed the bar for publication in Nature Communications.

Response: We appreciate your time for the review and supporting our manuscript for publication!